# Mechanisms leading to the 2016 giant twin glacier collapses, Aru Range, Tibet

Adrien Gilbert[1], Silvan Leinss[2], Jeffrey Kargel[3], Andreas Kääb[1], Tandong Yao[5], Simon Gascoin[6], Gregory Leonard[4], Etienne Berthier[7] and Alina Karki[8]

[1]Department of Geosciences, University of Oslo, Norway
[2]Institute of Environmental Engineering, ETH Zurich, Zurich, Switzerland
[3]Planetary Science Institute, Tucson, Arizona, USA
[4]Department of Planetary Sciences, University of Arizona, Tucson, USA
[5]ITP-CAS, Beijing, China
[6]CESBIO, CNES, CNRS, IRD, UPS, Université de Toulouse, Toulouse, France
[7]LEGOS, CNES, CNRS, IRD, UPS, Université de Toulouse, Toulouse, France
[8]Society for Ecological Restoration-Nepal, Kathmandu, Nepal

*Correspondence to*: Adrien Gilbert (adrien@geo.uio.no)

**Abstract.** In northwestern Tibet (34.0°N, 82.2°E) near lake Aru Co, the entire ablation areas of two glaciers (Aru-1 and Aru-2) suddenly collapsed on 17 July 2016 and 21 September 2016, respectively. The masses transformed into ice avalanches with volumes of 68 and 83•$10^6$ m$^3$ and ran out up to 7 km in horizontal distance, killing nine people. The only similar event currently documented is the 130•$10^6$ m$^3$ Kolka Glacier rock/ice avalanche of 2002 (Caucasus Mountains). Using climatic reanalysis, remote sensing, and three-dimensional thermo-mechanical modeling, we reconstructed in detail the Aru glaciers' thermal
regimes, thicknesses, velocities, basal shear stresses, and ice damage prior to the collapse. Thereby, we highlight the potential of using emergence velocities to constrain basal friction in mountain glacier models. We show that the frictional change leading to the Aru collapses occurred in the temperate areas of the polythermal glaciers and is not related to a rapid thawing of cold-based ice. The two glaciers experienced a similar stress transfer from predominant basal drag towards predominant lateral shearing in the detachment areas, and during the 5-6 years before the collapses. A high-friction patch is found under the Aru-
2 glacier tongue, but not under the Aru-1 glacier. This difference led to disparate behavior of both glaciers, making the development of the instability more visible for the Aru-1 glacier through enhanced crevassing and terminus advance over a longer period. In comparison, such signs were observable only over a few days to weeks (crevasses), or absent (advance) for the Aru-2 glacier. Field investigations reveal that those two glaciers were underlain by soft, highly erodible, and fine-grained sedimentary lithologies. We propose that specific bedrock lithology played a key role in the two Tibet and the Caucasus
Mountains giant glacier collapses documented to date by producing low bed roughness and large amounts of till, rich in clay/silt with a low friction angle. The twin 2016 Aru collapses would thus have been driven by a failing basal substrate linked to increasing water pore-pressure in the subglacial drainage system in response to increases in surface melting and rain preceding the collapse dates.

# 1. Introduction

In the Aru Mountain range on the western Tibetan Plateau the entire ablation zone of an unnamed glacier (termed here Aru-1) spontaneously collapsed on 17 July 2016. This occurred despite its low slope angle of only 13° (Tian et al., 2016) compared to typical ice avalanches occurring from the failure of much steeper hanging glaciers (Faillettaz et al., 2015). The Aru-1 glacier collapse produced an ice avalanche with speeds exceeding 200 km h$^{-1}$, spread over a 7 km long and 3 km wide deposit, and killed nine herders and hundreds of their animals (Kääb et al., 2018). This event was followed by the collapse of the adjacent glacier south of Aru-1 two months later, on 21 September 2016, producing a similar low-angle giant avalanche (see Figure 1B). Such catastrophic glacier collapses are extremely rare in size and mobility. Only one similar case has been documented before, the Kolka/Karmadon glacier collapse in the Caucasus Mountains in 2002 (Kääb et al., 2003; Huggel et al., 2005; Evans et al., 2009). In order to anticipate potential similar hazards in other populated mountain areas it is crucial to understand in detail the mechanisms involved, and identify potential triggers and factors responsible for these extreme mass movements. Among others, the collapses raise the question whether similar future events affecting other glaciers might be influenced by ongoing climate change.

Applying satellite image analysis and glacier mass balance modeling, Kääb et al. (2018) explored the long-term behaviour of the two Aru glaciers prior to collapse. They show that the two glaciers started a surge-like instability around 2010, probably in response to both increasing precipitation and temperature in the region, and related positive mass balances. Their preliminary analysis of the two-dimensional (2D) thermal glacier regime shows a polythermal structure for the two glaciers. Such a structure likely would have provided resisting forces against whole-glacier sliding, but would have promoted englacial drainage to the bed in the lower temperate part of the accumulation zone, with possible local sliding and contributing to swelling or inflation of the glacier toe above and behind the frozen part. Facing the enigma of two neighbouring glaciers undergoing—close in time—a similar catastrophic behaviour that otherwise is globally almost unique, Kääb et al. (2018) also point out the possible role that soft bedrock lithologies and glacier till production played in the instabilities.

In this study, we significantly extend the numerical analysis of the Aru glacier instabilities and discuss in detail the mechanisms leading to the collapses. We use a three-dimensional (3D) full-Stokes thermo-mechanical model in order to (i) reconstruct the bedrock topography, (ii) analyse in 3D the thermal regime of the glaciers, (iii) infer the evolution of basal friction prior to the collapse, and (iv) quantify the stress distribution that led to the final collapses. We then combine the modeling results with field investigations to further elaborate on the role of bedrock lithology and discuss the related origin of the twin collapses. Finally, we summarize key-characteristics, to recognize on other glaciers, lithologic and thermal regimes similar to the Aru glaciers to help identify new potential collapses in the future.

# 2. Observations

The Aru Range is located on the remote western Tibetan Plateau (34°N, 82°E) where very few glaciological or meteorological observations are available (Figure 1). Prior to Kääb et al. (2018), the two collapsed glaciers were never studied before and the

entire modeling work of this study is therefore based on remote sensing data and climatic reanalysis. Digital Elevation Model (DEM) differencing provided both the observations on the glacier transient dynamics and the mean mass balance over different time periods needed to calibrate the models. Kääb et al. (2018) compared different sources of local climatic data in order to reproduce remote-sensing based mass-balance observations and concluded that the ERA-interim reanalysis provides the best estimate of the Aru Range climate, if the respective precipitation amounts are corrected by a multiplying factor of about 4. Here, we use their mass balance model for constraining the thermo-mechanical model described in section 3.2.

## 2.1. Digital Elevation Models (DEMs)

We use seven different DEMs derived from different satellite missions between 2000 and 2016 (see Table 1). The SRTM C-band radar DEM from mid-February 2000 (Farr et al., 2007) is used as the steady-state reference of the two glaciers for reconstructing bedrock topography. A Pléiades optical satellite stereo DEM from 1 October 2016, after the collapse, allows us to evaluate the modeled bedrock reconstruction over the detachment zone. We compute ice emergence velocities by differencing pre-collapse high-resolution DEMs from TanDEM-X, Spot7 and WorldView data and correcting these for mass balance following the approach described in Gilbert et al. (2016) (Figure 2). The effect of uncertainty linked to radar penetration in the TanDEM-X data should be minimized when comparing same wavelength data (X-band) at similar times of the year. Change in penetration depth between the TanDEM-X data of 2011 (early June) and 2013 (mid April) due to different snow wetness should be also limited because surface melting in the accumulation area of the Aru glaciers only occurs from around mid-June on (Kääb et al., 2018). X-band penetration into glacier ice (i.e. the Aru glaciers ablation areas) is very limited anyway (Dehecq et al., 2016). Comparing Spot7 (2015) and TanDEM-X (2014) elevations likely introduces uncertainty from TanDEM-X penetration in the accumulation area leading to higher apparent emergence velocities in this part (visible in Figure 2). This problem only influences our friction reconstruction in the upper parts of the glaciers but not in the detachment areas. Details on DEM accuracies and acquisition methods can be found in Kääb et al. (2018).

## 2.2. Field observations

We investigated glacier till properties by analyzing samples collected from the Aru-1 avalanche deposit in the gorge close to the former glacier tongue. We collected these samples one year after the collapse on 18 July 2017. Rainy conditions on that day highlighted the behaviour of the surrounding lithology that quickly turned into a soft and unstable slurry in the presence of water. Additional information about our samples can be found in Figures S1-S3.

## 3. Modeling methods

### 3.1. Mass Balance

Our mass balance model for the two Aru glaciers is based on a degree-day model described in Gilbert et al. (2016). It has been calibrated for the Aru glaciers by using satellite-derived glacier mass balances and is fed by ERA-interim climate reanalysis

(Kääb et al., 2018). The model output, taken from Kääb et al. (2018), provides the spatio-temporal distribution of surface mass balance, firn thickness, and available surface melt water for percolation/refreezing in the firn to constrain the thermo-mechanical model below.

### 3.2. Thermo-mechanical model

Our thermo-mechanical ice-flow model is based on the Stokes equation coupled with an energy equation using the enthalpy formulation (Aschwanden et al., 2012; Gilbert et al., 2014). Changes in the glacier geometry are computed using a free surface equation (Gilbert et al., 2014). We adopt a pure viscous isotropic ice rheology following Glen's flow law (Cuffey and Paterson, 2010). The model is solved using the finite-element software Elmer/ice (Gagliardini et al., 2013). Parameters and variables of our model set-up are summarized in Table 2.

We adopt a linear friction law as a basal boundary condition for the Stokes equation:

$$\tau_b = \beta u_s \qquad (1)$$

where $\tau_b$ is the basal shear stress (MPa), $u_s$ is the sliding velocity (m yr$^{-1}$) and $\beta$ the friction coefficient (MPa yr m$^{-1}$). This

coefficient is inverted using a control inverse method to minimize a cost function defined from the misfit with measured surface data and a regularization term (Gillet-Chaulet et al., 2012; Gagliardini et al., 2013). Following Gilbert et al. (2016) we used the emergence velocity $U_{Nz}{}^{obs}$ (m yr$^{-1}$) to compute this cost function:

$$J_0 = \int_{\Gamma_s} \frac{1}{2}\left(\|U_{N_z}\| - \|U_{N_z}^{obs}\|\right)d\Gamma + \lambda J_{reg} \qquad (2)$$

$$J_{reg} = \frac{1}{2}\int_{\Gamma_b}\left(\left(\frac{\partial \beta}{\partial x}\right)^2 + \left(\frac{\partial \beta}{\partial y}\right)^2\right)d\Gamma \qquad (3)$$

where $U_{Nz} = (\mathbf{u}\cdot\mathbf{N})Nz$ is the modeled emergence velocity (m yr$^{-1}$), $\mathbf{u}$ the surface velocity vector (m yr$^{-1}$), $\mathbf{N}= (N_x,N_y,N_z)$ the unit vector normal to the surface, $\Gamma_s$ is the surface boundary, $J_{reg}$ is the regularization term, $\Gamma_b$ is the bedrock surface boundary and $\lambda$ is a positive number. The emergence velocity is obtained by removing the mean modeled mass balance from the elevation change rate measured from our repeat satellite-derived DEMs over the same periods (Figure 2):

$$U_{N_z}{}^{obs} = \frac{\partial h}{\partial t} - M \qquad (4)$$

where $\partial h/\partial t$ is the measured elevation change rate (m yr$^{-1}$) and $M$ the mean surface mass balance during the corresponding period (m yr$^{-1}$).

The surface boundary condition is set as a stress-free boundary for the Stokes problem and a Dirichlet condition for the enthalpy equation. In order to take into account water percolation and refreezing within the firn, we follow the approach by Gilbert et

al. (2015), using a 6-months time step. Latent heat due to refreezing is released every year during the summer time step. The

firn-thickness distribution is estimated from the mass balance model following Gilbert et al. (2016) and the firn density is computed using a linear density profile set to:

$$\rho(d) = \rho_0 + \frac{d}{H_{firn}}(\rho_{ice} - \rho_0) \quad (5)$$

where $\rho$ is the density (kg m$^{-3}$) at depth $d$ (m), $\rho_0$ is the surface density, $\rho_{ice}$ is the ice density and $H_{firn}$ the firn thickness (m). The lateral boundary condition is set to a no-flux condition for both the Stokes and enthalpy equations. We assume a basal heat flux of $8.0 \cdot 10^{-2}$ W m$^2$ for the enthalpy equation according to heat flux measured in boreholes at the Guliya ice cap (6200 m a.s.l., 200 km north of the Aru Range) (Thompson et al., 1995) and modelled geothermal heat fluxes for the region (Tao and Shen, 2008).

### 3.3.  Modeling strategy for the steady state glaciers

The first step of modeling the dynamics and thermal regime of the Aru glaciers is to establish a steady-state glacier as initial condition for 1970 (start of the climatic reanalysis used). Landsat satellite images of the glacier area and the mass balance model indicate that the two glaciers were close to equilibrium from 1970 to 1995 (Kääb et al., 2018). We therefore assume that the surface topography measured in February 2000 by the SRTM mission (oldest available DEM) is representative of the
glaciers being in equilibrium with the mean climate over this period, although the positive mass balance between 1995 and 2000 probably thickened the glacier by a few meters in the accumulation area. We use the mean mass balance between 1980 and 1995 as an equilibrium mass balance considering that modeled mass balance is close to steady-state during this period before becoming positive from 1995 to 2008 (Kääb et al., 2018).

We first run the model on a 2D flow line until a steady state is reached,  deriving bedrock topography in the detached parts
from a post-collapse Pléiades DEM, and by reconstructing the bed at the upper glacier parts assuming a constant basal shear stress (plastic approximation; Cuffey and Paterson (2010)). This initial step allows for the first approximation of the steady-state thermal regime which we presented in Kääb et al. (2018). We then use the 10 m-depth temperature modeled with the flow line model to define the steady-state surface enthalpy as a function of elevation which includes meltwater percolation and refreezing effects. This relationship is used to define a Dirichlet surface boundary condition for enthalpy in order to solve the
steady-state thermal regime of the glaciers in 3D in the bedrock inversion procedure (section 3.3.1). Because the effects of meltwater percolation and refreezing are already included in the surface enthalpy value, there is no need to solve for these effects in diagnostic runs. The final 3D steady-state glacier solution is obtained by running a transient simulation using the inverted bedrock topography and solving water percolation and refreezing until surface topography and the enthalpy field reach equilibrium with the imposed climatic condition.

### 3.3.1. Reconstructing bedrock topography

Using constant climatic conditions associated with the balanced glacier conditions corresponding to the SRTM DEM, we determined the bedrock topography allowing the best match between modeled and observed (i.e., SRTM DEM) surface topography (van Pelt et al., 2013). For this purpose, we ran a 3D transient simulation assuming no sliding, fixed surface topography (SRTM DEM), and constant surface forcing (mass balance and enthalpy). The no-sliding assumption is likely a good assumption in 2000 since the glacier was not surging at this time (Kääb et al., 2018). Mesh horizontal resolution is set to about 50 m with 15 vertical layers. The evolution of the free surface is taken into account by varying the basal mesh elevation instead of the surface elevation. The mesh surface topography remains thus constant while the bed topography is updated by solving the equation

$$\frac{\partial(z_{bed})}{\partial t} + \overrightarrow{v_s} \cdot \vec{\nabla}(z_{bed} - z_{bed0}) = \overrightarrow{v_s} \cdot \vec{\nabla}z_{s0} - M - w_s \quad (6)$$

where $z_{bed}$ is the bedrock elevation (m), $\overrightarrow{v_s}$ is the surface horizontal velocity (m yr$^{-1}$), $z_{bed0}$ is the initial bedrock topography (m), $M$ is the surface mass balance (m yr$^{-1}$), $w_s$ the vertical component of the surface velocity (m yr$^{-1}$) and $z_{s0}$ is the measured surface elevation (m). The right side of Eq. 6 vanishes once bedrock topography satisfies the required topography to keep $z_{s0}$ constant for a given mass balance $M$. The advective term in the left side of Eq. 6 allows smoothing $z_{bed}$ in the flow direction. The enthalpy field is solved at each time step by solving the steady-state equation for the current velocity field and mesh. We start with a uniform 200 m ice thickness (rough maximum expected thickness on the glacier) and run the model until reaching steady bedrock topography. This generates a new $z_{bed0}$ value to re-run the model until reaching a new steady state. After only two iterations, we validate the modeled bedrock topography by running the model with the new fixed bedrock topography and free surface evolution. The resulting surface topography is in excellent agreement with the measured one (Figure 3) indicating that our method to infer the bedrock topography works well for these two glaciers.

We use the opportunity provided by the exposed detachments to compare the reconstructed bedrock topography with the measured Pléiades DEM from after the collapses (Figure 3). On the Aru-2 glacier, the points where bedrock is clearly visible in the Pleiades images match well with the locations where our reconstructed bedrock topography matches the Pléiades DEM (dots in Figure 3). Elsewhere in the Aru-2 glacier detachment zone, the modeled bedrock is deeper than the observed surface elevation; this is likely due to the remaining ice debris overlying the actual bedrock, so the Pléiades DEM elevations are expectedly higher. This is confirmed by the good continuity between the ground topography measured outside of the former glacier tongue and the one inferred from our bedrock reconstruction (see Figure 3, profile 6). On the Aru-1 glacier, the reconstructed bedrock on profiles 2, 3 and 4 is systematically deeper than the Pléiades DEM, even on the steep side close to the margin of the glacier where no ice remained after the collapse. This means that ice flow is not accurately modeled in this part, likely due to the premise of no sliding, which is probably not accurate considering that the glacier may have been temperate at its base here (see section 4.1). The error in the modeled bed topography of the Aru-1 glacier is, however, $< 30$ m

and will only slightly affect the absolute value of the friction coefficient inferred during the instability development (see section 4.2), but not its relative changes, which are the focus of this study. The assumption of no sliding should also affect the result on the Aru-2 glacier, which has a similar thermal regime, but where the no-sliding condition seems to work. This indicates the existence of different sliding conditions for the two glaciers prior to collapse, which is also supported by the friction inversion analysis presented in section 4.2. In the upper parts of both glaciers, the no-sliding assumption is, however, supported by the friction inversion analysis.

## 4. Results

### 4.1. Steady state configuration of the two glaciers

The Aru glaciers are representative of a cold and semi-arid climate regime, and thus would normally show little dynamic behaviour under mostly cold-ice conditions (below the pressure melting point). The steady-state equilibrium line is located around 5750 m a.s.l. (minimum glacier elevations around 5200 m a.s.l., and maximum elevations around 6100 m a.s.l.) with a maximum accumulation of 0.6 m w.eq. yr$^{-1}$ at 6100 m a.s.l. and a maximum ablation of -2.5 m w.eq. yr$^{-1}$ on the tongue (Figure 4B). Both glaciers are composed of two similar catchments characterized by a smaller western branch that joins the main stream in the ablation area. The western branch of each glacier is thinner and less dynamic than the main branch that collapsed in summer 2016 (Figure 4A). Maximum surface horizontal velocity reached 20 m yr$^{-1}$ in the Aru-2 glacier, which have to accommodate higher ice flux than the Aru-1 glacier due to a wider accumulation area (1.7 km$^2$ vs 1.2 km$^2$) converging in a similarly narrow gorge.

As previously concluded by Kääb et al. (2018), our results show that the main branches of the two glaciers are characterized by a polythermal structure with a cold accumulation zone above 5900 m a.s.l. and a temperate-based ablation area surrounded by cold ice (Figure 5). However, through the more accurate bedrock topography derived in this study and the 3D approach, we show here that the temperate zones likely extended into significantly larger areas beneath the detachments than previously thought. Temperate ice forms in the lower part of accumulation zones due to a significant amount of percolation and refreezing of melt water, which increases the temperature of the near-surface firn. This warmer ice is then advected into the ablation zone contributing, together with basal heat flux, to create temperate basal conditions in the lower parts of the two glaciers. Cold surface conditions due to absence of water percolation in the ablation zone (cold impermeable ice) lead to a significant cold surface layer that eventually reaches the glacier base in the shallowest zones of the glacier tongues (Figure 5). The western branches of the two glaciers have a significantly smaller temperate area with an ablation zone almost entirely cold-based (Figure 5). This thermal structure may explain why the western branches remained stable after the collapses even though each branch lost its downstream supporting buttress formed by the detached glacier tongues. The modeled spatial extent of the temperate basal ice, under steady-state conditions coincides with the detached areas and indicates that friction changes leading to the collapse occurred in temperate ice rather than being produced by a change from cold to temperate thermal conditions at the glacier beds. However, the large amount of cold ice, especially along the side of the gorge, could have provided significant

lateral drag that built up high driving stress, which was able to balance gravitational force under the frictional change at the temperate parts of the beds.

### 4.2. Basal friction change since 2011

The surge-like behaviour of the two glaciers identified from DEM comparison in Kääb et al. (2018) documents a change in the glacier dynamics during the five years prior to the twin collapses. By removing the elevation change due to surface mass balance we quantify the emergence velocity for constraining the basal friction parameter (Gilbert et al., 2016) for different periods: 2011-2013, 2013-2014, 2014-2015 and September-November 2015 (Table 1, Figure 2). Our results highlight a contrasting behaviour between the Aru-1 and Aru-2 glaciers where friction decreased progressively in magnitude through time in both glaciers, but over significantly different areas (Figure 6). Frictional changes over the five years prior to collapse are also more significant on the Aru-1 glacier, resulting in a higher increase in surface velocity than on the Aru-2 glacier (Figure 7). Similarly inferred friction for the Aru-2 glacier for annual means (2011-2013 and 2013-2014) and a 2-month mean (Sept-Nov 2015) indicates low seasonal variability of the basal condition. Similarly, modeled surface velocities on the Aru-1 glacier in Sept-Nov 2015 correlate well with those measured for Jan-Apr 2016 using satellite image correlation (Kääb et al., 2018) (Figure 7F), also indicating low seasonal variability.

### 4.3. Force balance analysis

To evaluate how resisting forces acted and evolved to balance the driving forces, we compute the mean basal shear stress during different periods from the inverse method. We therefore assume that basal shear stress is mainly constrained by the global stress balance and should not be influenced by the sliding law that we used (eq. 1) (Joughin et al., 2004; Minchew et al., 2016). The steady-state condition shows a basal shear stress between 100 kPa and 200 kPa in both glaciers with mean shear stresses of 137 kPa and 150 kPa for the Aru-1 and Aru-2 glaciers, respectively (Figure 8A). In comparison, mean driving stresses are 152 kPa (Aru-1) and 213 kPa (Aru-2) indicating that 10% (Aru-1) and 30% (Aru-2) of the driving force is accommodated by normal force along the sidewalls. These levels of driving stress are at the higher end of the observed range of driving stresses on mountain glaciers (Cuffey and Paterson, 2010) and reflect the presence of strong resisting forces due mainly to cold-ice conditions combined with the resistance of the valley walls.

The inversion of mean elevation changes between September and November 2015 (Figure 8B) reveals that basal shear stresses on the Aru-1 glacier decreased to only 20 to 10 kPa in large areas, and basal resistance became mainly achieved by a few sticky spots (Stokes et al., 2007) in the detachment zone where shear stresses exceeded 250 kPa. Along the left bank of the glacier, close to the terminus of the Aru-1 glacier, shear stress was about 6-7 kPa and was not more than 15 kPa at the terminus. In comparison, the Aru-2 glacier behaved differently with more localized friction changes that produced a smaller change in the distribution of the basal shear stress during the same period (Figure 8C).

The analyses of the dynamics and force-balance evolution on an area restricted to the detachment zone (dashed red lines in Figure 8) reveals both similarities and differences between the two events (Figure 9). Further references below to "lateral

stress" apply to the detachment zone and not to the whole glacier, and refer to the stress provided by the shearing interface between the stable and the unstable part of the glacier (visible in Figure 9D). On the one hand, as already highlighted in Figure 7, the mean detachment velocity prior to collapse behaved differently for the two glaciers (Figure 9A). While the Aru-1 glacier detachment significantly accelerated, following behaviour typical for slope failure (Voight, 1990) over several years (blue dashed line in Figure 9A), the Aru-2 glacier showed very little acceleration. On the other hand, force balances evolved similarly in the two detachments with a large increase of lateral stresses along the detachment margin due to both an increase in the driving stress  and reduction in basal friction (Figure 9B). Interestingly, lateral resistance overcomes basal resistance in both detachments with a delay time (81 days) close to the actual delay between the two final collapses (66 days) (Figure 9C). On the Aru-2 glacier, it seems that smaller changes in friction were compensated by a higher change in driving stresses resulting in a similar increase of stress at the detachment margin compared to the Aru-1 glacier (Figure 9B). The difference in surface velocity response to these similar stress transfers was a consequence of different basal drag repartitions in the two glaciers. Basal drag decreased uniformly on the whole detachment of the Aru-1 glacier with the appearance of localized sticky spots, whereas basal drag decreased only in the higher part of the detachment of the Aru-2 glacier. This led to more intense bulging and a lower velocity increase (Kääb et al., 2018) due to the high-friction patch remaining in the tongue (Figure 6).

To evaluate the impact of the friction change on the mechanical property of the ice, we compute the maximal principal Cauchy stress and compare it with a threshold value set to 0.1 MPa (Krug et al., 2014) to identify the damage production (crevasse opening) (Krug et al., 2014; Pralong and Funk, 2005) (Figure 10). The modelled stress fields clearly highlight zones where a progressive intensification of cracks opened around the detachment zone of the Aru-1 glacier (Figure 10C) as observed on satellite images (Kääb et al., 2018); these fractures led to its final collapse. In comparison, the Aru-2 glacier again behaves differently with less damage (cracks) that only affect the upper part of the detachment (Figure 10C). This means that  damage at the shear margin would have occurred suddenly in the Aru-2 glacier in 2016, which is confirmed by the observed sharp crack surrounding the detachment only a few days before the collapse (Kääb et al., 2018). In sum, the Aru-1 and Aru-2 glaciers underwent similar stress transfers, transitioning from basal drag to lateral shearing in their respective detachments, but showed different responses in terms of damage (i.e. crack production) and sliding speed due to different basal drag repartition. the Aru-1 glacier progressively evolved towards collapse whereas the Aru-2 glacier accumulated stresses until a sudden release led to collapse. This indicates that critical stress transfers, precursory to  such collapses, may occur without observable phenomena (i.e., surface velocity increase, crevassing) in the preceding years.

## 5. Discussion

### 5.1. Result uncertainties

The modeled thermal regime is sensitive to basal heat flux, which is poorly constrained. However, sensitivity tests (see supplementary material, Figure S4) show that the temperate area remains stable for a basal heat flux between $6.0 \cdot 10^{-2}$ and $1.2 \cdot 10^{-1}$ W m$^{-2}$ and disappears only at $\leq 2.0 \cdot 10^{-2}$ W m$^{-2}$. Measurements in the Guliya Ice Cap (Thompson et al., 1995) and

reconstructions from Tao and Shen (2008) both give a value of $8.0 \cdot 10^{-2}$ W m$^{-2}$, making a low value of $\leq 2.0 \cdot 10^{-2}$ W m$^{-2}$ very unlikely. Kääb et al. (2018) have also shown that firn thickness has a great influence on the modelled thermal regime around 5900 m a.s.l. where melting occurs in the accumulation zone. Firn thickness is, however, hard to estimate without field investigation; following Kääb et al. (2018), we applied an intermediate scenario where firn thickness linearly increases from

the ELA to the glacier top where it reaches a 15 m maximum. Sensitivity test showed that only very little firn thickness (< 5m at 6000 m a.s.l.) would lead to an almost cold glacier (supplementary material, Figure S5). Nevertheless, the modeled thermal regime and the friction reconstruction, which are both almost independent from each other, are in good agreement with the localization of sliding and modeled temperate areas, lending confidence in our results despite uncertainties in basal heat flux and surface boundary conditions (see section 5.2).

The uncertainty in the reconstruction of basal friction mainly depends on the accuracy of measured elevation changes, which is generally higher over longer time periods (increased signal-to-noise ratio), making the 2011-2013 reconstruction the most reliable one. The measured Sept-Nov 2015 elevation change is subject to a lower signal-to-noise ratio and is thus poorly resolved in the accumulation area (see Figure 2). However, it leaves the reconstruction reliable on the detachment area where elevation changes are much more significant. A similar conclusion applies to the 2014-2015 reconstruction where the upper

part of the glacier is affected by penetration of the X-band signal, leading to an overestimation of the emergence velocities (see Section 2.1). However, the influence of this uncertainty on the modeled mass balance used to compute emergence velocity is also low in the detachment zone, since elevation changes due to surface mass balance are relatively small compared to the dynamical height changes linked to the surge-like instability (<20%) (Figure S6). Our results are, therefore, least affected by uncertainties and most reliable in the detachment area, which is the focus of this study. In addition, bedrock topography is well

constrained in the detachment areas from the post-collapse Pléiades DEM, giving additional confidence in the friction reconstruction there.

Using emergence velocities to constrain basal friction is not a commonly used method, and has been successfully tested only on a slow-flowing ice cap by Gilbert et al. (2016). We therefore provide additional validation of this method in the supplementary material (Figures S7-S9) by inverting the friction on both glaciers using horizontal velocities inferred from

offset tracking obtained from repeat TerraSAR-X data in December 2013. This test reveals good agreement between our emergence-based approach and the more common method based on horizontal velocities. In particular, sliding zones are similarly localized in both methods. Using the inversion based on horizontal velocities as a reference, we estimate a sliding speed magnitude accuracy of 0.036 m day$^{-1}$ in the emergence-based inversion. Our additional validation test also indicates that using emergence velocities may provide for an improved constraint of the friction coefficient in accumulation areas. The

reason for this is that the underlying data used in generating the emergence velocities (i.e. DEMs, modeled mass balance) are often more spatially resolved and cover larger areas on small mountain glaciers, as opposed to measurements of horizontal displacements, which have problems over accumulation areas.

## 5.2. Frictional changes

Our results indicate that low friction below the Aru glaciers was not linked to seasonal variability of water pressure, which is often observed on glaciers elsewhere (Bartholomaus et al., 2008; Vincent and Moreau, 2016). Rather, it is likely associated with sustained change of the basal conditions caused by an accumulation of liquid water over several years prior to the collapse. Over a hard bed (Cuffey and Paterson, 2010), this would likely result in the existence of a subglacial lake which is very unlikely here because the low friction zone on the Aru-1 glacier extended to the tongue and the lake should have drained in such case. Futhermore, in temperate ice, high water pressure conditions are unstable over long time periods because they lead to channel formation that can efficiently drain water and decrease the pressure (Schoof, 2010). High water pressure in a cavity network would be also difficult to maintain in the Aru cases since increasing sliding speeds tend to increase cavity size and decrease water pressure. These arguments suggest that basal friction under the Aru glaciers was probably controlled by processes associated with soft bed properties (Cuffey and Paterson, 2010).

Comparison between sliding speed evolution and modeled basal steady-state temperature reveals a good correlation between the zones of sliding and temperate ice conditions and shows that the size of sliding areas remains similar over time (Figure 11). This confirms that friction reduction since 2011 mainly occurred within zones that were already temperate areas, and that friction reduction is therefore not linked to a simple change from cold to temperate basal conditions. However, contrary to the Aru-1 glacier, Aru-2 appears to have been affected by a high-friction zone under its lower tongue, which the modeled basal temperatures are not able to explain as they indicate temperate, not cold conditions (Figure 11). This zone of high friction explains the different behaviours observed between the two glaciers in terms of surface velocities and glacier advance. Indeed, a few months before the collapse, the Aru-2 glacier velocities were still low compared to the Aru-1 glacier (Figure 7) and while the Aru-1 glacier advanced almost 200 m since July 2015, the front of the Aru-2 glacier remained stationary until the collapse (Kääb et al., 2018). Although the high-friction zone may have delayed the collapse of the Aru-2 glacier, it did not prevent it.

## 5.3. Role of the bedrock lithology and glacier till

Field observations after collapse and inspection of the detachment zones showed no presence of a hard-bed lithology beneath the glaciers, and no large boulders were observed in the forefields and avalanche deposits. Rather, extensive deposits of soft, unconsolidated and fine-grained lithologies were identified (Figures S1-S3). We collected till samples from the Aru-1 glacier avalanche deposit and measured their grain-size distribution (Figure 12). Mean values over the four samples in the avalanche path (Fig. 12) indicate till consisting of 14% clay, 24% silt, 44% sand and 18% gravel. These samples are representative of the material found in the deposit and are likely also representative of the glacier till on which the glacier rested. We also observed a rather smooth-surfaced failure interface (i.e. detachment plane) suggesting a low bedrock roughness at the macro scale (>1m). These findings confirm that the Aru glaciers rested on a soft bed, which likely played an important role in controlling the behaviour of the glaciers from the surge initiation to the collapse. For such bed types, basal motion is not controlled by ice

flow around bedrock bumps (Weertman, 1964; Lliboutry, 1968) but rather by deformation in till (Truffer et al., 2001). The sustained very low basal drag under the Aru glaciers (<20 kPa) may be similar to ice stream mechanisms whereby water-saturated till enables fast flow at low driving stresses (≈20 kPa) (Cuffey and Paterson, 2010). It has been shown that glacier till behaves with a plastic rheology with a shear strength strongly dependent on the effective normal stress (Iverson et al., 1998; Clarke, 2005; Iverson, 2010). The use of a linear friction law in our inversion can be viewed as a parametrization where β includes these physics and is only valid at the time of the inversion. Therefore, the change in friction coefficient β can be interpreted in terms of a plastic till. Such behaviour was found to be well described by a Coulomb-type friction law (Boulton and Jones, 1979; Clarke, 2005) as follows:

$$\tau_u = c + N\tan(\phi) \quad (7)$$

where $\tau_u$ is the ultimate shear strength, $c$ the cohesion parameter, $N$ the normal effective stress and φ the friction angle. This kind of law, where shear stress is independent of the sliding velocity, allows for unstable behaviour leading to failure. In a general case, glaciers remain stable because till and water pressure are not equally distributed at the basal interface, leading to sticky spots where stress concentrates to balance gravitational forces together with lateral drag at glacier margins (Cuffey and Paterson, 2010). The Aru collapses would thus be an example where basal shear stress becomes limited to till strength in such large zones that resisting forces can no longer balance gravity, eventually leading to catastrophic failure. The latter would happen for a bedrock with low roughness, which provide less vestigial resistance (surface-normal stresses) to constrain ice velocity in case of failure in the till. We propose that the change in effective normal stress due to increasing pore pressure in the till beneath the temperate zones of the Aru glaciers quickly reduced the ultimate strength of glacier till and limited the basal shear stress to a maximum value lower than the driving stress. The glacier shape could not adjust fast enough to reduce the driving stress due to strain-rate limitation in the cold-based zones, leading then to the accumulation of large stresses in the remaining sticky spots (Figure 8), until their sudden rupture. The sticky spots were likely remnants of stiff frozen till rather than solid rock irregularities, rendering them susceptible to failure under high stress and vulnerable to thaw from water-saturated temperate surroundings and increasing deformational heat. The high clay and silt content measured in the till is indicative of lithologies having unique low friction-angle properties (Iverson et al., 1998) and higher sensitivity of the shear strength to changes in water pressure.

One likely scenario for the development of the now collapsed Aru glaciers is that they grew in the past (pre-industrial climate) in colder conditions with low melting rates in summer, allowing for rigidity in the basal till to support high driving stress (see section 4.3). The low water pressure meant that there was likely very little sliding and therefor very little production of till at that time. Upon commencement of some sliding, which may have occurred gradually over an increasing area of the bed during the past century, till production increased and the local glacier deformation regime tended to adapt to the distribution of till and liquid water reaching the bed. At this stage, the percolation into, and accumulation of meltwater beneath the glaciers increased so rapidly in recent years (Kääb et al., 2018) that the glaciers could not keep balance with the changing conditions

at the bed. Sliding may also have contributed to increase the water pressure in a positive feedback by destroying any efficient drainage system (Clarke et al., 1984). The contributory role of soft-bed lithology in the collapses is therefore likely threefold by (i) behaving with plastic rheology when shear strength is reached, (ii) providing for low roughness at the ice-bed contact, and (iii) maintaining high water pressure while sliding speed increases; a known process that accounts for surge behaviour

(Clarke et al., 1984; Raymond, 1987; Fowler et al., 2001). High content in clay and silt probably also leads to low hydraulic conductivity favourable to higher water pressure in the till (Fowler et al., 2001).

We suggest that the existence of a high friction area under the Aru-2 glacier tongue prior to collapse is due to both higher basal normal stresses (Figure S10) which increased the till strength, and higher lateral drag along the west side of the detachment which decreased the basal shear stresses compared to the Aru-1 glacier (Figure 9). In this way, and contrary to the Aru-1

glacier, basal shear stress under the tongue of the Aru-2 glacier only approached the ultimate shear strength of the till just before the final collapse in response to both decreasing resistance by the lateral margin (due to crevassing) and increasing driving stresses (due to bulging).

### 5.4. Till-strength controlled glacier collapses

The Aru collapses, and in retrospect the Kolka Glacier collapse, define a newly recognized type of avalanching glaciers

characterized by an underlying failing substrate. Such "iceslides" could occur on glaciers with fairly low angle, involving therefore potentially large volumes of material, and presenting serious consequences in terms of hazard potentials. The high sensitivity of the ultimate shear strength of the substrate to pore water pressure, combined with low bed roughness, allows for a dramatic and sustained change of basal friction conditions capable of driving this kind of instability.

The Kolka event in 2002 in the Caucasus Mountains is probably another example of this type of instability in which the

maximum shear strength of the till is exceeded by a sudden increase of basal shear stress at constant effective normal stress. Indeed, during the few weeks before this collapse, significant mass was added on top of the glacier by rock and ice-fall activity increasing basal shear and normal stresses (Huggel et al., 2005; Evans et al., 2009). This could change the surface slope, which reached the till friction angle and triggered the failure. If the till was saturated with water and had low hydraulic conductivity, increasing pore water pressure could also have compensated for the rising normal stress keeping the normal effective stress

constant. Once maximum shear strength is exceeded, failure is triggered (Evans et al., 2009). This hypothesis is plausible since Kolka Glacier is known to be a surging glacier, able to store large amounts of liquid water, and high water content and pressure were observed before its 2002 collapse (e.g. unusual ponds observed on the glacier prior to collapse) (Kotlyakov et al., 2004). However, changes in till-strength in response to changing in water pressure are likely also involved in the surge mechanisms of temperate glaciers without the large majority of surges turning into gigantic collapses. This renders sudden changes in till

strength as a necessary, but not a fully sufficient condition, for collapses controlled by till-strength. The necessary secondary conditions for catastrophic failure is a sustained high driving stress with low bed roughness, coincident with weakening till. This means that the glacier has to grow over time, atop of a more stable substrate capable of supporting higher driving stresses. In particular, freezing conditions allow for the development of relatively thick glaciers on slopes that would otherwise be

unable to support such high shear stress under the presence of liquid water. This makes the spatio-temporal interplay of soft-bed characteristics and the polythermal glacier regime a prerequisite of the Aru collapses, whereas for the Kolka Glacier the additional loading over a short time should have caused a fast increase in shear stress significantly exceeding the glacier's normal conditions.

In many of the world's glacierized regions, on-going atmospheric warming increases surface melt and the amount of water reaching glacier beds, thereby modifying the till shear strength. This development is therefore in theory capable of driving more till-strength controlled instabilities and collapses. The most impacted glaciers would be those flowing on soft and highly erodible bed lithologies at high driving stress, particularly those with heterogeneous thermal structure (polythermal glaciers). Such glaciers are mostly localized in cold and dry climates where a small increase in temperature results in a relatively large

change in melting conditions such that the amount of water reaching the glacier base can significantly increase instability. In reality, however, an array of factors and their specific (and to this point rare) interplay in space and time are necessary to catalyse glacier collapsing as observed for the Aru and Kolka glaciers.

## 6. Conclusion

In summer 2016, one of the most spectacular glacier disasters ever observed occurred in western Tibet. The collapses of the

twin Aru glaciers set a new reference in terms of size and mobility of glacier instabilities, and required a reassessment of assumptions and conditions that more typically drive hazards and impacts linked to mountain glaciers. Using 3D thermo-mechanical modelling together with satellite and field observations we conclude that the Aru twin collapses were driven by increasing melt water reaching the bed in the temperate area of the polythermal structure of the glaciers, leading to the weakening of the underlying till and sediment .

Our steady-state simulation reveals that both glaciers were likely polythermal, with predominant temperate basal conditions over the detachment areas. Using satellite-observed elevation change and modeled surface mass balance, we reconstructed the frictional and shear stress regimes at the glacier base that occurred during the five years prior to collapse. We show that both glaciers experienced a stress transfer in their detachment area, transitioning from basal drag to lateral shearing at the detachment margin, likely beginning around 2012. However, the different spatial repartitions of basal drag in the two

detachment zones led to visibly different behaviours. As early as 2015, basal drag in the Aru-1 glacier was very low over the whole detachment zone with a few remnant sticky spots where stress was concentrated. In contrast, basal drag of the southern Aru-2 glacier was distributed between a low-friction area in the upper half of the detachment zone and a high-friction area in the lower half. These circumstances led to a progressive destabilization of the Aru-1 glacier with a significant acceleration in ice flow in the detachment zone over several years prior to collapse, whereas stresses accumulated in Aru-2 until a sudden

break of the shear margin occurred only a few days before the collapse.

We interpret that the change in friction was due to glacier till reaching its ultimate shear strength in response to increasing pore water pressure. This assumption is supported by field observations that revealed soft and erodible material with high clay/silt

content underneath the glaciers. Plastic rheology of the till underlying the Aru glaciers combined with low bedrock roughness and polythermal glacier structure seem to be the basis of the collapses. The polythermal structure enabled the glaciers to grow at high driving stress on a partially frozen substrate while temperate areas facilitated the water to reach the bed. Increasing water pressure in temperate areas led to failure in the till and thereby to increasing shear stresses on localized sticky spots and along the detachment margin. Due to the low bed roughness, the nature of these sticking spots seems purely thermal (cold patches). They are therefore mechanically susceptible to failure and can be affected by thermal effects such as intense deformational heat or latent heat release.

Under climatic changes and related increase in surface melt rates, polythermal glaciers underlain by soft and erodible substrate are likely to destabilize more readily than hard-bed glaciers. Lower bed roughness of the former and plastic rheology of such till  promotes instability, while hydrological feedbacks with high till shear rate destroying efficient drainage components (canals) leads to increasing pore water pressure and weakening  substrate strength. The Aru Glaciers cases highlight the most extreme of glacier behaviours when bedrock roughness and/or frozen zones are unable to sustain global stability while the substrate is failing, leading to the catastrophic failure of large glacier sections.

## Acknowledgements

We would like to thank Irina Rogozhina and Martin Truffer for their valuable and constructive review comments. We are further grateful to the satellite data providers: CNES for Pléiades, Airbus/CNES for Spot 7, DLR for TanDEM-X, and Digital Globe for WorldView. A.K. and A.G. acknowledge the Univ. Oslo EarthFlows initiative and funding from the European Research Council under the European Union's Seventh Framework Programme (FP/2007-2013)/ERC grant agreement no. 320816, and A.K. also acknowledges the ESA projects Glaciers_cci (4000109873/14/I-NB) and DUE GlobPermafrost (4000116196/15/IN-B). S.G. and E.B. acknowledges support from the French Space Agency (CNES) and the Programme National de Télédétection Spatiale grant PNTS-2016-01. A.G. acknowledges the Institute of Tibetan Plateau for organizing a field trip in summer 2017, and M. S. Naoroz at the Department of Geosciences, University of Oslo, for his support with the sample analyses. J.K. thanks NASA (High Mountain Asia Team) for support.

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

30

**Table 1 – Different DEMs used in this study produced from satellite data**

| Satellite /Sensor | Acquisition date | DEM resolution | Image type |
|---|---|---|---|
| **SRTM C/X** | February 2000 | 30 m | Radar |
| **TanDEM-X** | 6 June 2011 | 10 m | Radar |
| **TanDEM-X** | 14 April 2013 | 10 m | Radar |
| **TanDEM-X** | 01 April 2014 | 10 m | Radar |
| **Spot7** | 06 September 2015 | 5 m | Optical |
| **WorldView** | 25 November 2015 | 5 m | Optical |
| **Pléiades** | 01 October 2016 | 5 m | Optical |

5  **Table 2 – Variables and parameters of the thermo-mechanical model.**

| Name | Symbol | Value | Unit |
|---|---|---|---|
| Velocity | $\boldsymbol{u}$ | - | m yr$^{-1}$ |
| Stress tensor | $\boldsymbol{\sigma}$ | - | MPa |
| Pressure | $P$ | - | MPa |
| Enthalpy | $H$ | - | J kg$^{-1}$ |
| Temperature | $T$ | - | K |
| Water Content | $\omega$ | - | - |
| Density | $\rho$ | - | kg m$^{-3}$ |
| Firn thickness | $H_{firn}$ | - | m |
| Friction coefficient | $\beta$ | - | MPa yr m$^{-1}$ |
| Flow Rate factor | $A$ | f$(T)$[a] | MPa yr$^{-1}$ |
| Glenn's law exponent | $n$ | 3 | - |
| Basal heat flux | $f_b$ | 0.080 | W m$^{-2}$ |
| Thermal conductivity | $k$ | f$(\rho)$[b] | W K$^{-1}$ m$^{-1}$ |
| Heat Capacity | $C_p$ | f$(T)$[c] | J kg$^{-1}$ K$^{-1}$ |
| Maximum water content | $\omega_{max}$ | 0.03 | - |
| Moisture diffusitvity | $\kappa_0$ | 1.045 10$^{-4}$ | kg m$^{-1}$ s$^{-1}$ |
| Residual saturation in firn | $S_r$ | 0.01$^{3}$ | - |
| Firn surface density | $\rho_0$ | 350 | kg m$^{-3}$ |

[a](Cuffey and Paterson, 2010) ; [b](Calonne et al., 2011) ; [c](Gilbert et al., 2014)

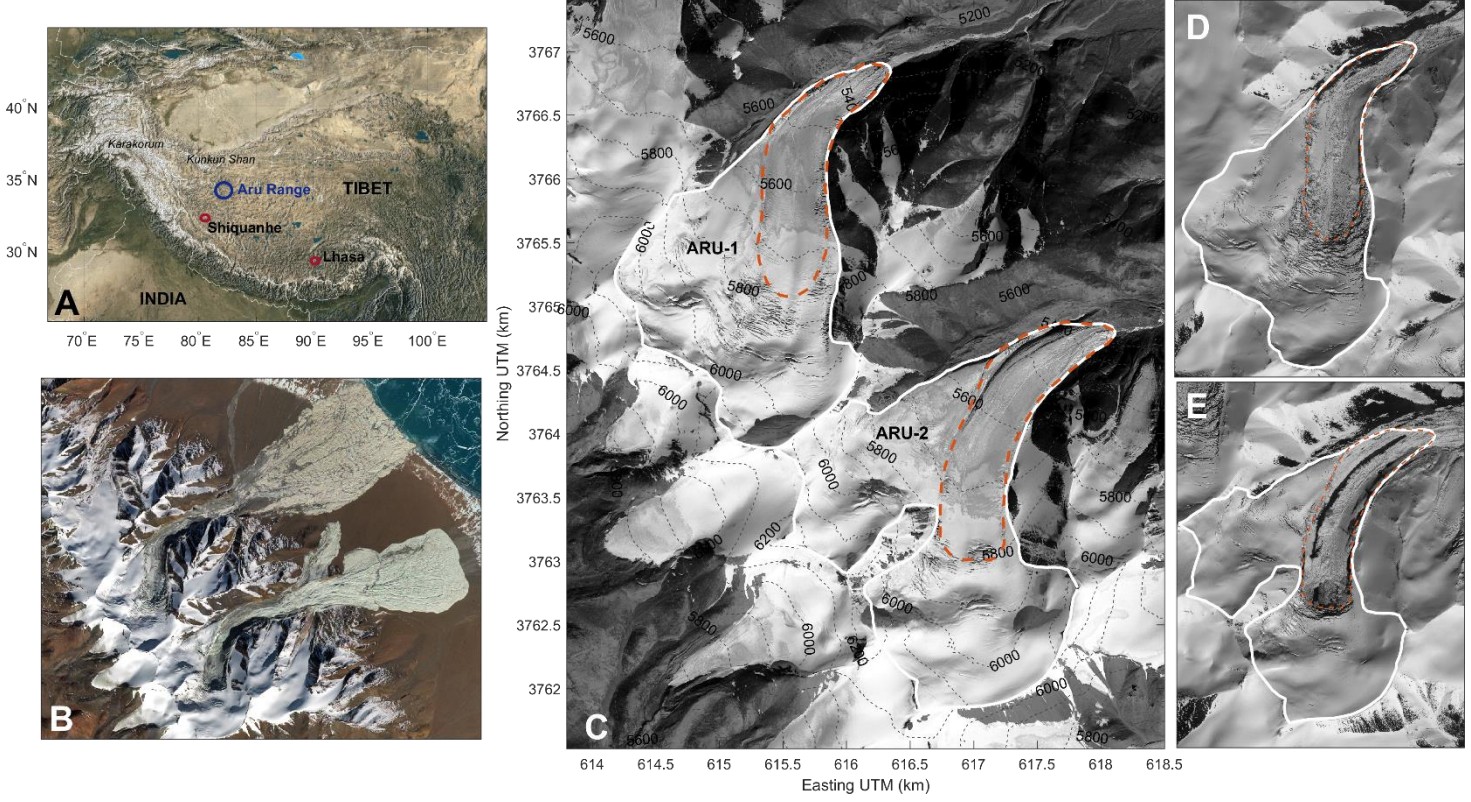

**Figure 1** – **(A)** Location of the Aru Range in Tibet and **(B)** Sentinel-2 image from 2016, December 8 after the collapses (Contains modified Copernicus Sentinel data 2016). **(C)** Elevation contour lines of Aru-1 and Aru-2 glaciers (vertical datum WGS84) overlaid on an orthorectified Spot7 image from 2015 (Copyright Airbus D&S), September 21 as background used also in all other figures unless otherwise stated; orange dashed lines indicate the detachment outline, white lines are the glacier outline as of 2015. **(D, E)** Pléiades images from 2016, October 1 of the two glaciers Aru-1 **(D)** and Aru-2 **(E)** after the collapses (Copyright CNES 2016, Distribution Airbus D&S).

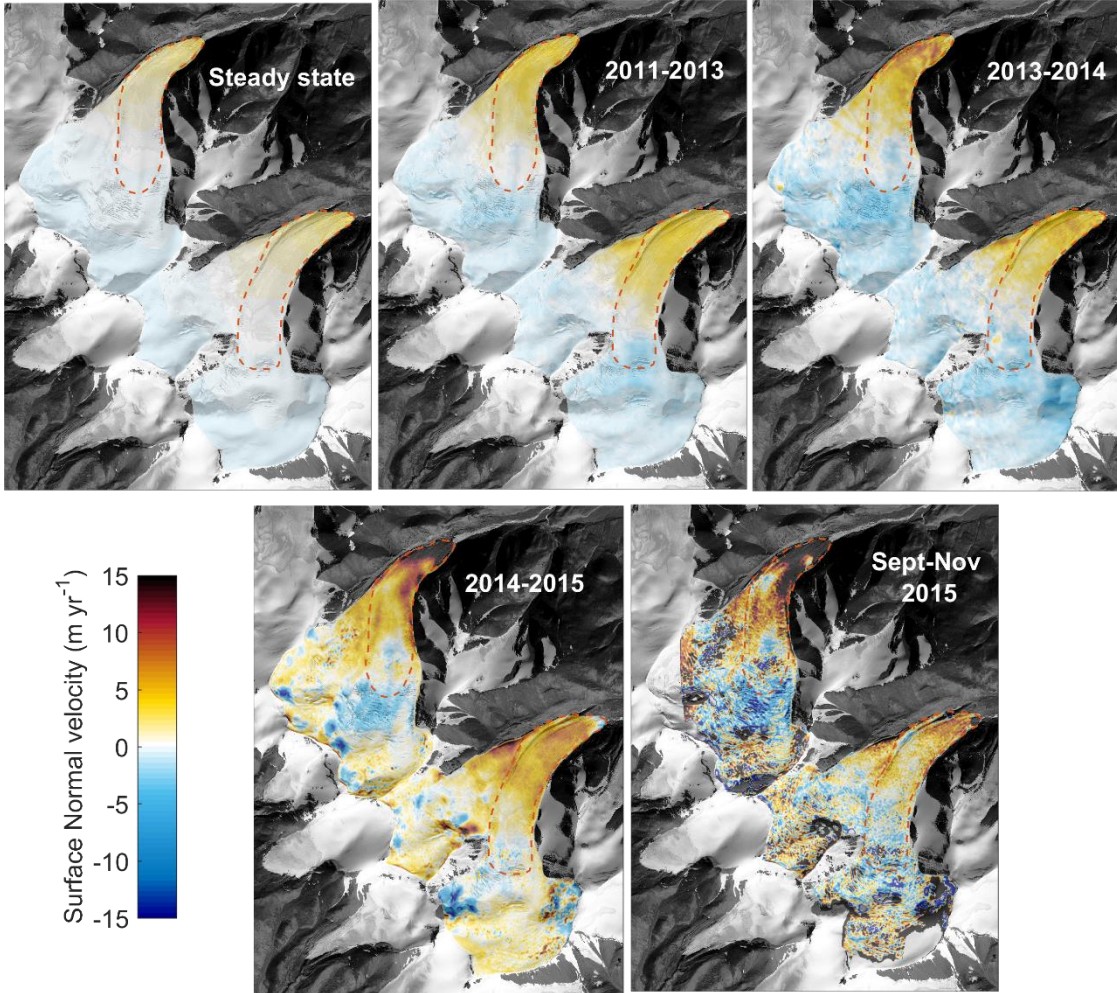

**Figure 2 – Mean emergence velocities obtained by differencing elevation changes from repeat DEMs and modeled mass balances during different periods prior to the collapses. Steady-state velocities in the first panel are modeled. Orange dashed lines indicate the detachment outline.**

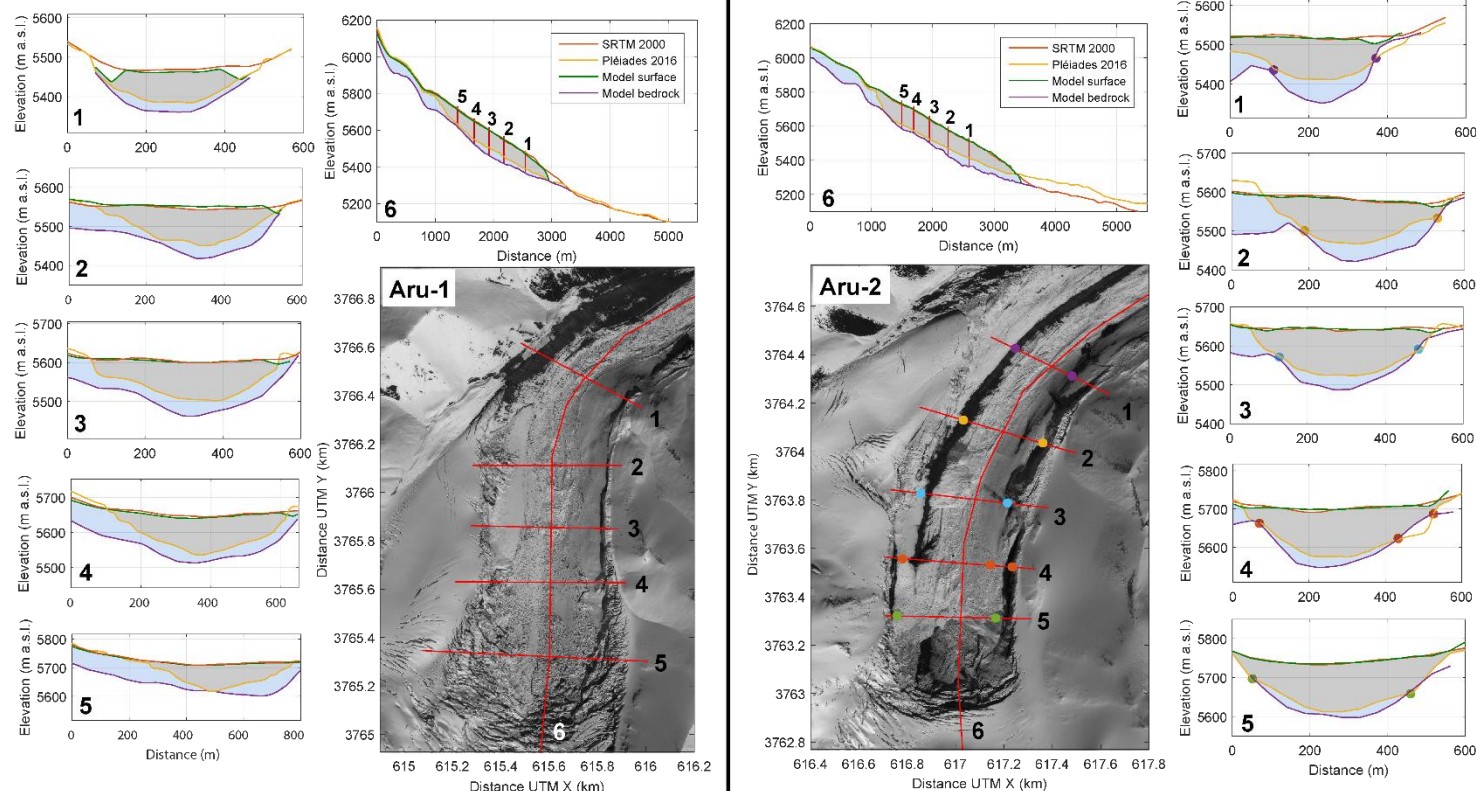

**Figure 3 – Pléiades image of Aru-1 (left) and Aru-2 (right) glaciers after collapse with topographic profiles 1 to 6 plotted for both glaciers (Copyright CNES 2016, Distribution Airbus D&S). The topographic profiles 1 to 6 show the measured surface topography in 2000 (SRTM, in red) and 2016 after the collapse (Pléiades, in yellow). These profiles are compared with the modeled bedrock (in purple) and surface (in green) topographies. The colored dots on the Aru-2 glacier show the location of specific points of the profiles in the Pléiades image. Those points correspond to locations where our reconstruction matches the Pléiades DEM and where bedrock should thus be visible on the Pléiades image (no ice debris). Grey shading indicates the detached parts according to the Pléiades DEM compared to SRTM.**

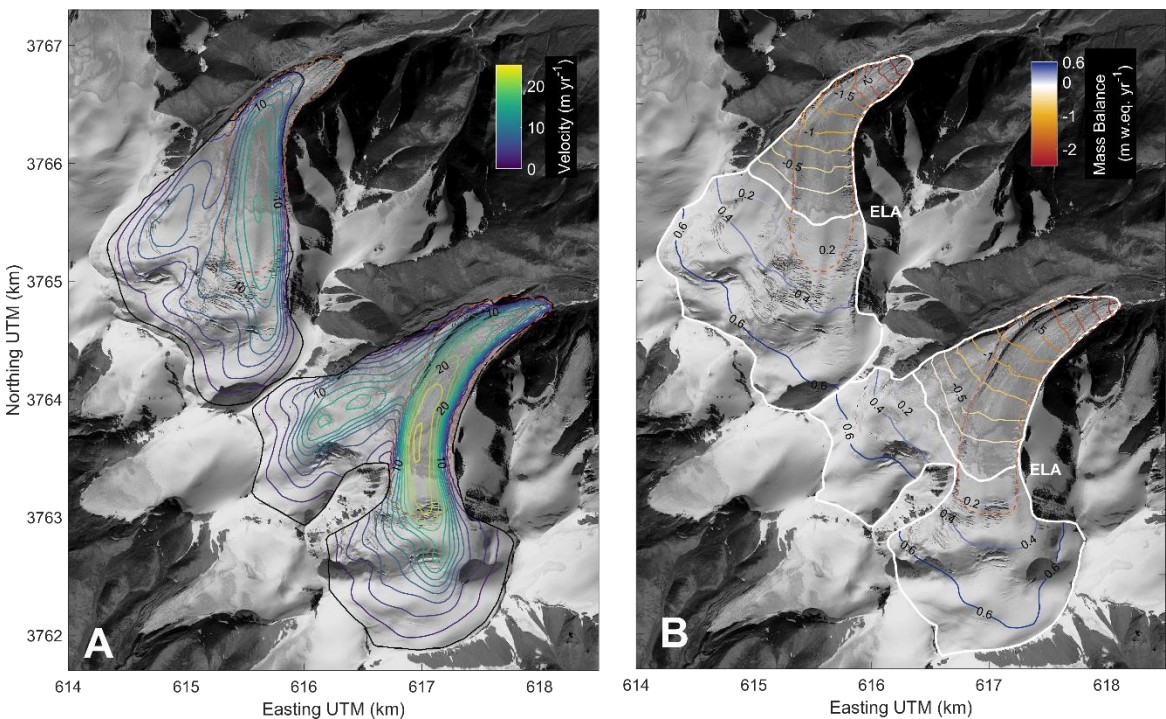

**Figure 4 – Modeled steady-state horizontal surface velocities (A) and surface mass balance (B) for Aru-1 and Aru-2 glaciers. The black contours in (A) are modeled steady-state glacier outlines. The white contour in (B) is the glacier outline as mapped from 2015 images. Orange dashed lines indicate the detachment outline.**

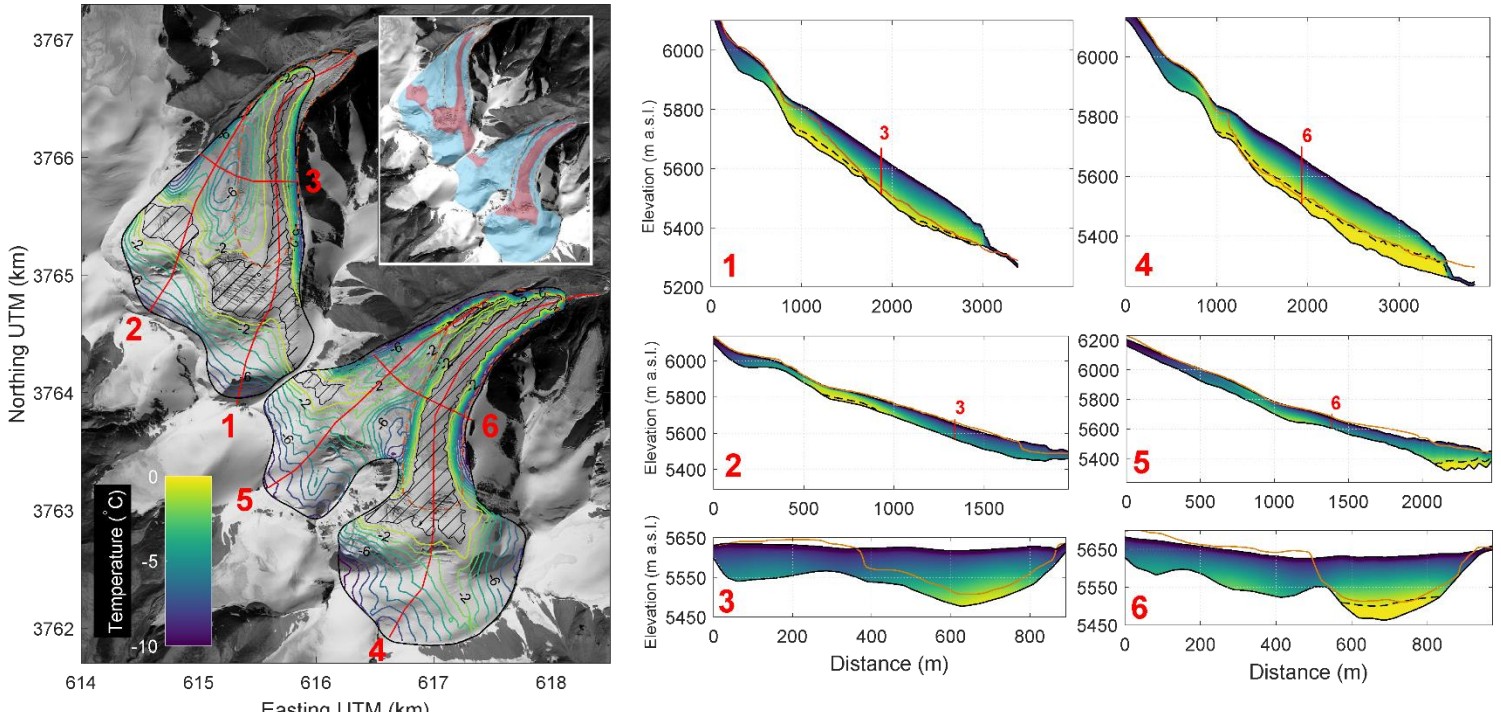

**Figure 5 - Modeled steady state temperature on the Aru-1 and Aru-2 glaciers. Left panel shows basal temperature with black hatched lines showing temperate areas. The inset highlights temperate-based (red) and cold-based (blue) areas. Orange dashed lines indicate the detachment outline. Right panels show 2D temperature profiles 1 to 6 as indicated in the left panel (red lines). Profiles include Pléiades 2016 elevation profiles (orange lines). The dashed black lines indicate the cold-temperate transition surface. Note that vertical scale is exaggerated in profiles 1 and 4.**

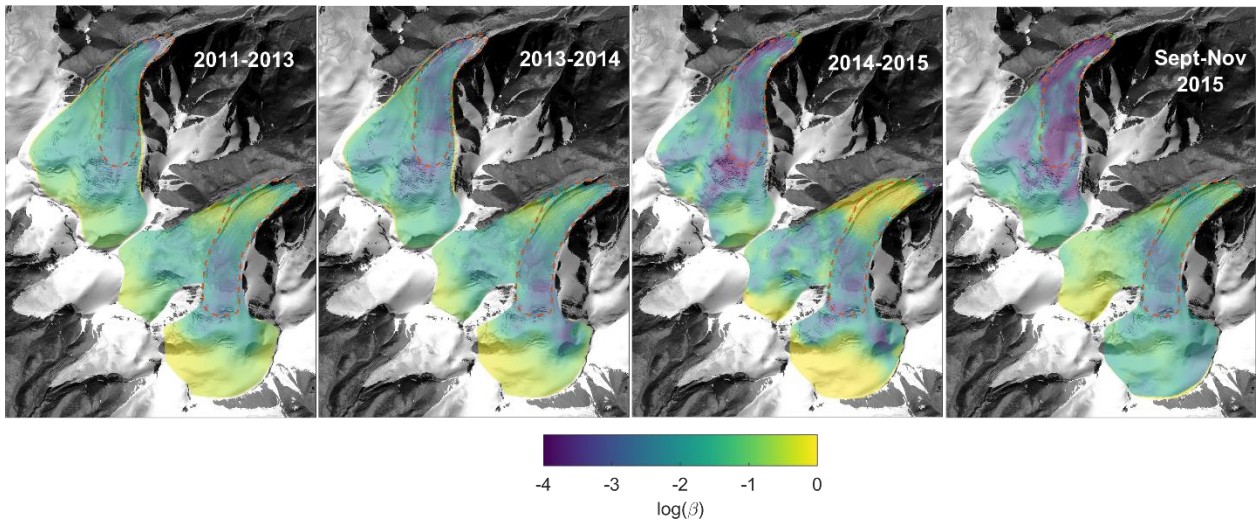

**Figure 6 - Friction coefficient β inferred from emergence velocity during different periods prior to the collapse. Orange dashed lines indicate the detachment outline.**

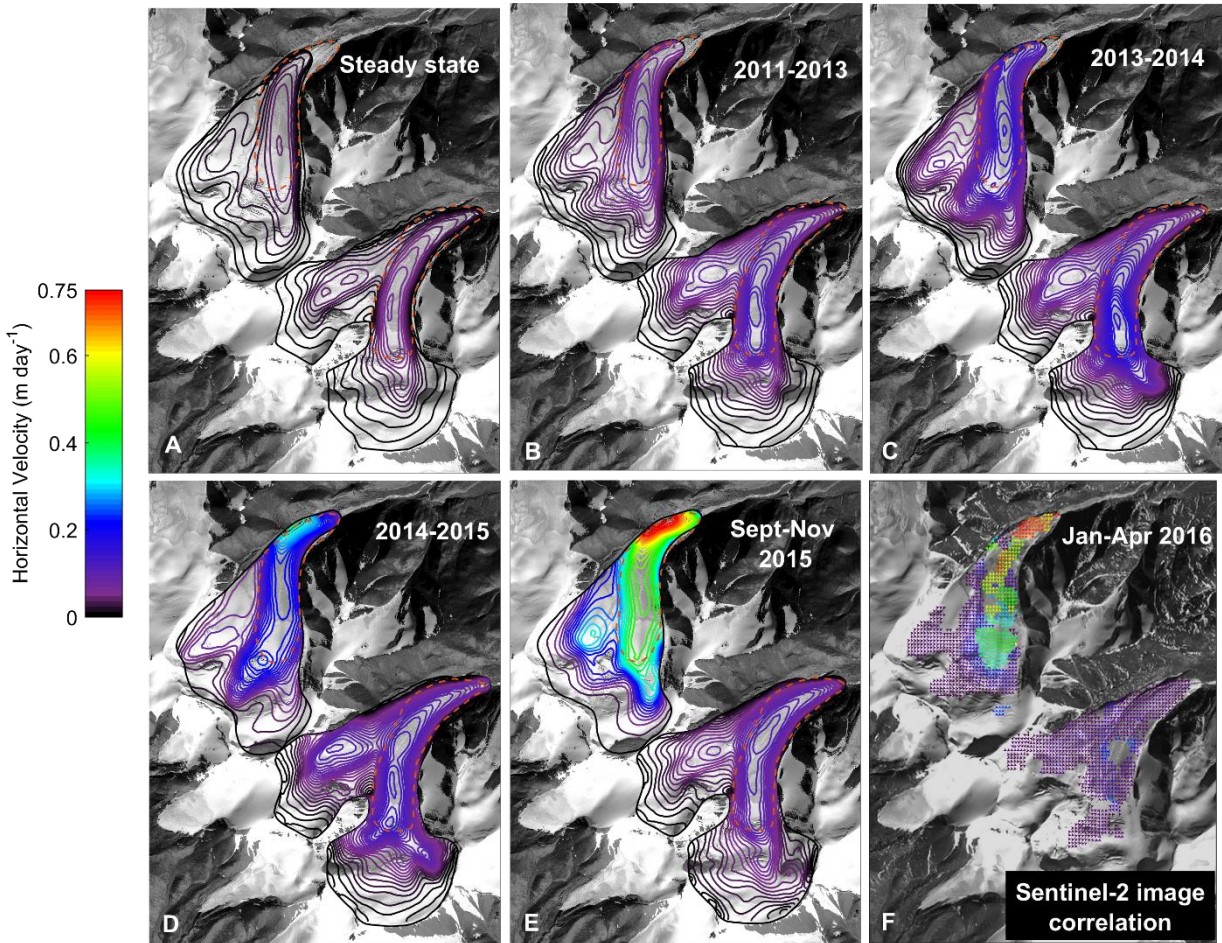

**Figure 7 – (A-E) Modeled mean horizontal velocities for the two glaciers at steady state and for the periods 2011-2013, 2013-2014, 2014-2015 and September to November 2015. (F) Measured horizontal velocities from Sentinel-2 image correlation between January and April 2016 (Adapted from Kääb et al. (2018)). Orange dashed lines indicate the detachment outline.**

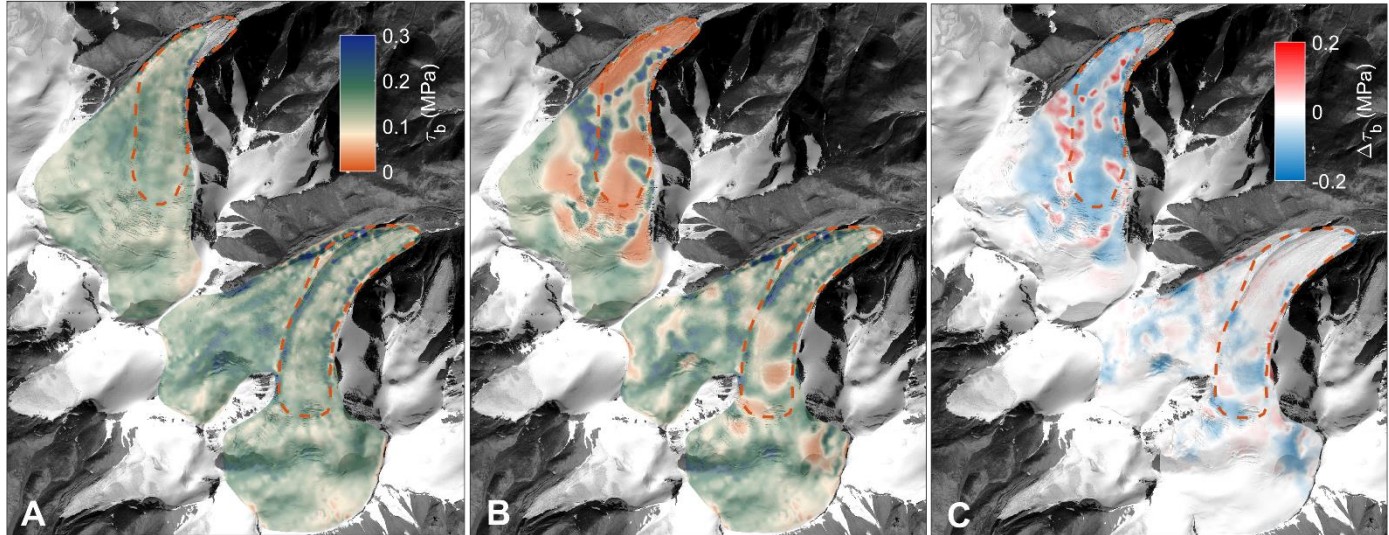

**Figure 8 – Modeled basal shear stress at steady state (A) and in November 2015 (B). (C) is the difference between (B) and (A).**

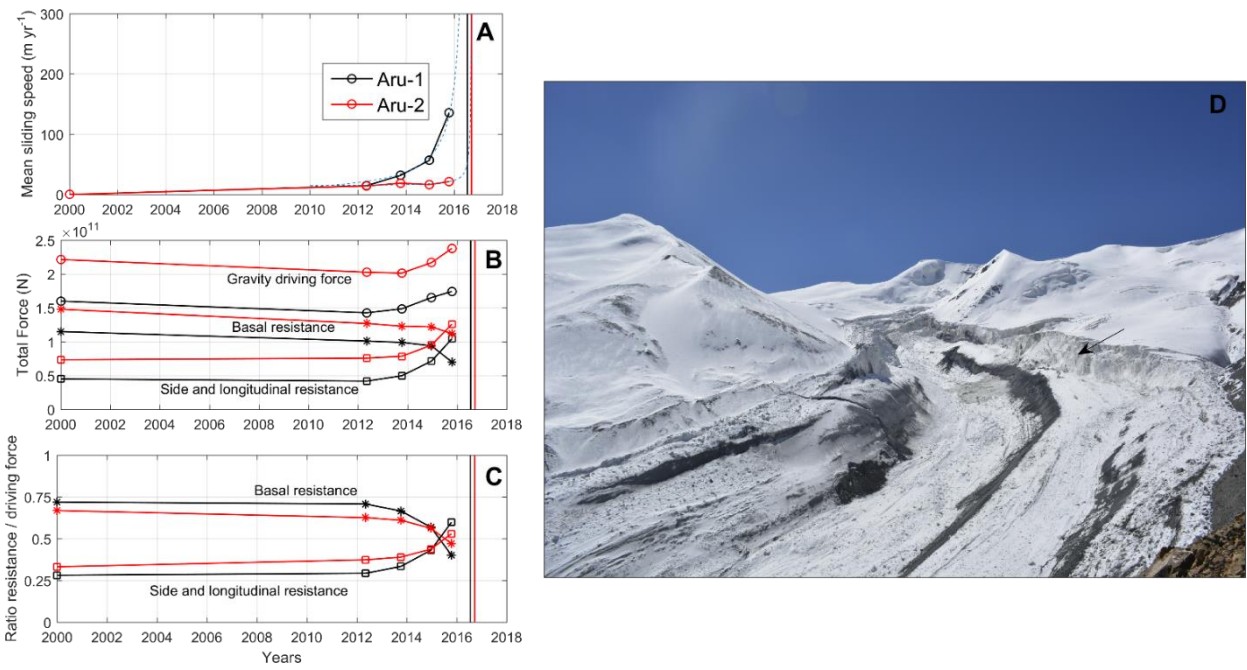

**Figure 9 – (A) Mean sliding speed of the detachment zone for Aru-1 (black) and Aru-2 (red) glaciers. The dashed blue lines show predicted speed following an empirical law of slope failure (Voight, 1990). (B) Force balance of the detachment along the glacier bed direction. (C) Ratio of resisting force over driving force. Vertical lines show collapse dates in the three panels. (D) The Aru-2 glacier detachment zone on 4 October 2016 (Picture from T. Yao). The side resistance computed in the force balance analysis arise from lateral shearing at the detachment right side (black arrow).**

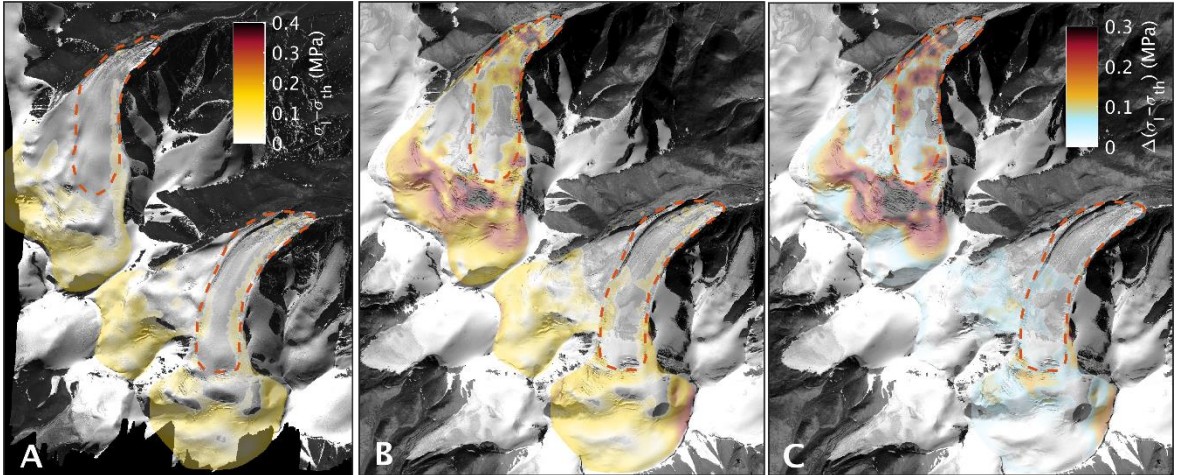

**Figure 10 – Maximum principal Cauchy stress excess above damage initiation threshold at steady state (A) and prior to collapse (B) at the glacier surface. (C) shows the difference between (A) and (B). Background image in (A) is a WorldView image from 2011, December 2 when the instability just started. Background image in (B) and (C) is a Spot 7 image from 2015 September 21, one year before collapse (copyright Airbus D&S). These results show a good match between predicted and observed crevasse formation in response to frictional changes.**

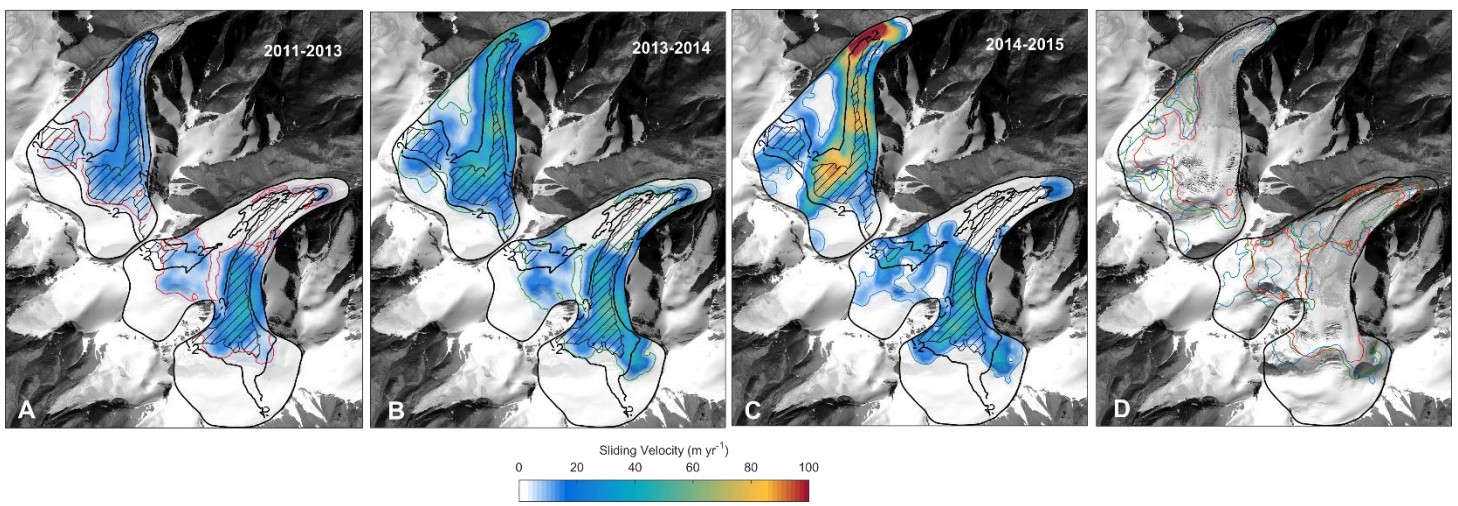

**Figure 11 – (A, B, C) Modeled temperate area (hatched zones) and -2 °C isotherm at steady state compared with sliding speeds over different periods (background colormap). (D) Comparison of sliding location for the different periods.**

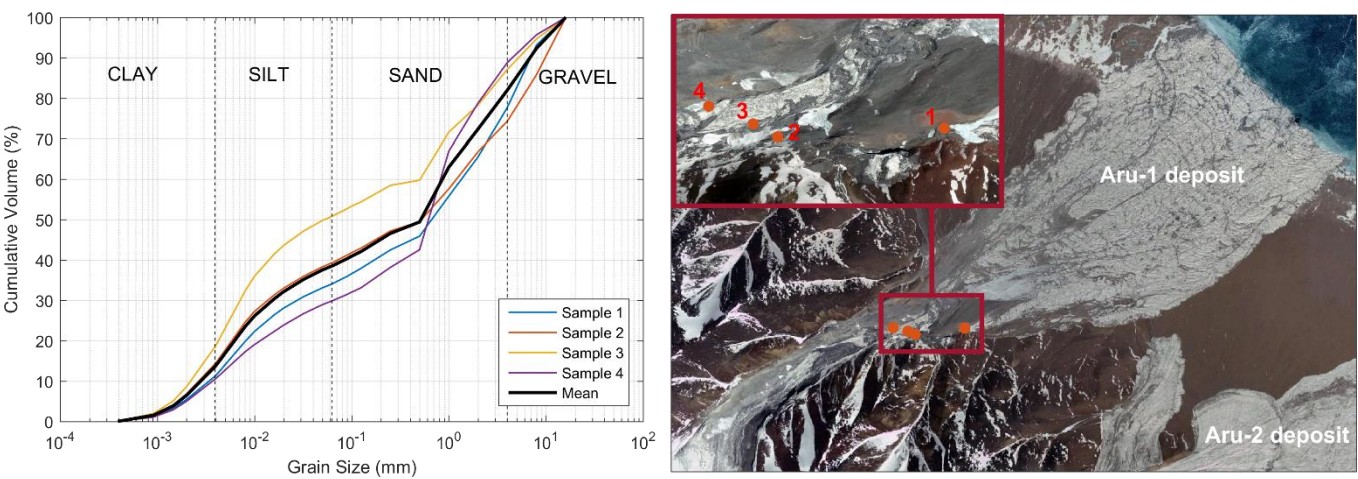

**Figure 12 – Grain-size distribution measured in four glacier till samples collected in the Aru-1 deposit area (numbers 1 to 4 in the right panel). Background image Google Earth.**