# Peer review of "Mechanisms leading to the 2016 giant twin glacier collapses, Aru"

_The Cryosphere, 2018_

## Referee Comment (RC1) · M. Truffer (Referee) · 28 May 2018

**General Comment**

This paper discusses the catastrophic collapse of two glaciers in Tibet within a few months of each other; one of the most astonishing glaciological events ever recorded. The paper provides a thorough analysis of the glaciers' development in the years prior to the event using satellite data and climate models interpreted with a thermomechanical 3D ice sheet model. The paper reaches substantial conclusions that might even be a bit counter-intuitive (i.e. it is not just melting of a previously frozen bed). It should be published after some modification. Most importantly, it needs thorough editing. There are many grammatical mistakes including long convoluted sentences, missing

pronouns, misused prepositions, confusing singular and plural, and third person singular.

Specific Comments

1) The method for deriving basal friction is not well explained. It is not very common to use vertical velocities for the cost function in an inversion. There is a reference to Gilbert et al (2016), but that paper uses both vertical and horizontal velocities, which is likely to constrain the friction parameter much better. If I understand correctly, vertical velocities are derived from DEM differences (yielding dh/dt) and a mass balance model? If so, why is there a discussion of surface-normal velocities? Both dh/dt and b_dot are generally evaluated in the vertical direction, so there is no need for this? Intuitively, I'm surprised that this method works so well, but the results do look encouraging. But there should at least be some discussion of errors (which is missing for any of the results).

2) There are conflicting assumptions in the paper that are not always discussed. For example, the derivation of bed topography is based on 'no sliding' (this is shortly discussed). Friction parameters are derived from a linear sliding law, but the discussion is entirely in terms of a plastic till.

3) I would like a bit more information about how stresses are divided between basal shear stress and lateral stress. In a valley geometry, the bed-parallel stress can be both lateral and basal.

4) The Kolka Glacier case is interesting with a rock fall on it. There is a simple argument to be made that for a plastic till the addition of a mass on top of the glacier will lead towards instability if the glacier slope is larger than the friction angle of the till, without invoking pore water pressure changes. Is this potentially the case here?

5) The abstract mentions that this is a response to recent increases of surface melting and rain. Neither is shown in the paper. This is an important conclusion and only

enters the paper via a mass balance model that is discussed elsewhere. For such a substantiative statement it seems like there needs to be at least some amount of backup (e.g. a figure of temperature/precip changes)

Technical corrections

I won't list grammatical issues, there are too many. This paper needs a very careful editorial revision. Some other comments:

p.2,l.7: unique -> rare (it's not unique you mention another example in the next sentence...)

p.3,l.10: are the two X-band images from the same time of year? Otherwise could the penetration depth change with snow wetness?

eqn (2): d should by y

p.6,l.2: which two cases?

p.7,l.31/32: I don't understand that sentence at all (.. external side of the curve ..)

p.10,l.2: sec 5.2 is a self-reference...

p.10,l.10: How did you observe bedrock roughness. I thought this was all till covered?

p.10,l.17/18: The MacAyeal and Tsai references don't quite seem appropriate here; they don't show plastic till, they assume it in their models.

p.10, last paragraph: I find some of the discussion here confusing. What do you mean when you state that 'plastic rheology becomes the only source of resisting forces'? Or 'increasing pore pressure ... quickly reduced basal shear stress'? Increasing pore water pressure reduces effective stress (not shear stress) and through that the strength of the till. In a plastic rheology you can't reduce the shear stress to the strength; till strength is a limiting stress.

p.11,l.17: What would cause higher lateral stresses? See also my earlier comment:

when does a basal stress become a lateral stress in a valley geometry?

---

## Referee Comment (RC2) · I. Rogozhina (Referee) · 3 Jun 2018

Reviewer: Irina Rogozhina, MARUM, University of Bremen/NTNU

This paper presents a model-based interpretation of temporal changes in the internal dynamics, basal friction and stress states of two glaciers in western Tibet to explain their catastrophic collapses in 2016. The inversion method used to derive glacier model results is rather unusual – in this I agree with reviewer 1 – but it seems to yield rather good results. I still think that this method should be validated on a glacier that has measurements of horizontal velocities, vertical surface changes and ideally bedrock topography to make the case that it is operational. There are quite some examples of such glaciers, especially in the European Alps. Inversion methods can be quite tricky,

since they derive whatever one wants to obtain, especially when multiple parameters are estimated in parallel. Nevertheless, this study is an impressive contribution to the state-of-the-art understanding of the glacier dynamics and addresses the challenge of the glacier model initialization in a neat manner, even though it has a significant overlap in terms of motivation and conclusions with the paper featuring the same authors (Kääb et al., 2018). The language is quite remarkable, as reviewer 1 has pointed out, and I am rather surprised to see so many experienced co-authors – including native speakers – who do not seem to have read the paper. With this review I encourage them to have a look at it. I believe that this paper will merit publication in TC after moderate revisions.

MAJOR POINTS: 1. As I mentioned in my summary, the authors should prove that their inversion method is operational by validating it on a glacier with more measurements (see above). 2. I absolutely agree with reviewer 1: All the points he has raised are valid and I am looking forward to seeing responses to his concerns. In addition, I feel that his specific point 2 needs further exploration: It would be worth looking at how friction angle in equation 4 (in addition to friction coefficients) changes over time leading to the collapses of the glaciers. In addition to showing how the subglacial till changed its properties in response to warming and increased meltwater supply, this experiment will provide an estimate of the yield stress needed to enable such a failure. A very useful exercise for the future diagnostic experiments that will empower predictions of similar glacier failures and an important exercise to support the conclusions of this study. 3. I don't believe much in the climate forcing provided by ERA-Interim in such high-topography, steep-gradient environment, especially after I learned from this paper that the precipitation rate had to be multiplied by a factor of 4. Could the authors compare ERA-interim fields with High Asia Refined (HAR) analysis (Maussion et al., 2014)? I suggest that the authors perform sensitivity tests to assess the uncertainties in their results coming from the mass balance estimates using HAR.

MINOR POINTS: The methods section is sloppy. For example, I am missing a table with model parameters. In general, the methods have to be more detailed. This is

not a Nature paper, there is space for the description of methods. The supplementary materials are no accessible through the online system. There are some citations of materials in the supplement, which I cannot access. Page 7, lines 13 – 14: Cannot it be influenced by a larger error in the bedrock estimate? Page 9, line 9 – 13: This requires a proof. Page 9, line 27: "from temperate to cold basal conditions" - the other way around? Page 10, section 5.3: The field data are only available for Aru 1. Are the authors sure that Aru 2's bed has the same lithology? Page 11, lines 4 – 5: Any evidence from the little ice age glacial moraines to support this statement?

References: Kääb, A., Leinss, S., Gilbert, A., Bühler, Y., Gascoin, S., Evans, S. G., Bartelt, P., Berthier, E., Brun, F., Chao, W.-A., Farinotti, D., Gimbert, F., Guo, W., Huggel, C., Kargel, J. S., Leonard, G. J., Tian, L., Treichler, D. and Yao, T.: Massive collapse of two glaciers in western Tibet in 2016 after surge-like instability, Nat. Geosci., 11(2), 114–120, doi:10.1038/s41561-017-0039- 7, 2018. Maussion, F., D. Scherer, T. Mölg, E. Collier, J. Curio, and R. Finkelnburg: Precipitation seasonality and variability over the Tibetan Plateau as resolved by the High Asia Reanalysis, J. Climate, 27, 1910-1927, doi:10.1175/JCLI-D-13-00282.1, 2014.

---

## Author Comment (AC1) · 2 Jul 2018

**Mechanisms leading to the 2016 giant twin glacier collapses, Aru range, Tibet**

Adrien Gilbert et al.

**Response to reviewer comments**

We would like to sincerely thank the referees for their careful feedback on our study that certainly helped to improve its presentation. We believe we can sufficiently respond to all comments made and improved the manuscript accordingly

Reviewer comments in normal font.
Response in *italic blue* font

- **Reviewer 1 (M. Truffer)**

**General Comment**

This paper discusses the catastrophic collapse of two glaciers in Tibet within a few months of each other; one of the most astonishing glaciological events ever recorded. The paper provides a thorough analysis of the glaciers' development in the years prior to the event using satellite data and climate models interpreted with a thermomechanical 3D ice sheet model. The paper reaches substantial conclusions that might even be a bit counter-intuitive (i.e. it is not just melting of a previously frozen bed). It should be published after some modification. Most importantly, it needs thorough editing. There are many grammatical mistakes including long convoluted sentences, missing pronouns, misused prepositions, confusing singular and plural, and third person singular.

*We have now significantly improved the grammar throughout the manuscript and believe its language is now good enough for the final language editing done by the publisher.*

**Specific Comments**

1- The method for deriving basal friction is not well explained. It is not very common to use vertical velocities for the cost function in an inversion. There is a reference to Gilbert et al (2016), but that paper uses both vertical and horizontal velocities, which is likely to constrain the friction parameter much better.

*Gilbert et al. (2016) do validate the method by providing both velocity field inferred from emergence velocity only (figure 8A of their manuscript) and from combination of emergence and horizontal velocities (figure 8C of their manuscript). They show this method is working well. We provide now new additional results based on horizontal velocity measurements from TerraSAR-X offset tracking between 2013-11-30 and 2013-12-11 that give a new validation of the method. As discussed in the manuscript we do not see signs of strong seasonal variability and these results should be similar to the ones obtained by TanDEM-X DEM differences between 2013-04-14 and 2014-04-01. Comparison between these two results shows a good accordance in the reconstruction of surface velocities (Figure R1) which validates our method. Furthermore, horizontal velocities obtained after friction inversion from DEM differences between 2015-09-06 and 2015-11-25 match well the observed horizontal velocity between January and April 2016 providing another validation of our method (see Figure 7 of the manuscript). The detailed comparison of modeled sliding velocities between the two methods shows that the localization of sliding zone are in reasonable agreement despite some differences in magnitude (Figure R2). In particular, sliding velocities are underestimated in the upper half of the Aru-1 detachment zone. This highlight a lack of accuracy in the emergence-velocity-based method that show a standard deviation of 13.1 m yr$^{-1}$ (0.036 m day$^{-1}$) compared to the horizontal velocity inversion (Figure R2-D).*

[Figure]

*Figure R1 – Surface horizontal velocity modeled after basal friction inferred from horizontal velocity measurement (A) and from emergence velocity estimation (B). (C) Measured horizontal velocity between 2013-11-30 and 2013-12-11 from TerraSAR-X offset tracking.*

[Figure]

*Figure R2 - Sliding horizontal velocity modeled after basal friction coefficient inferred from horizontal velocity measurement (A) and from emergence velocity estimation (B). (C) shows difference between (A) and (B) and (D) is the associated error distribution (blue) and Gaussian fit (red).*

*However, due to not completely spatially resolved horizontal velocity measurements, especially in slow flowing glacier zones (see Figure R1-C), the friction coefficient is better-constrained using emergence velocity in those zones. This leads to significant difference in those regions between the two reconstructions in terms of friction coefficient (Figure R3). We believe that our method based on emergence velocities is a powerful method for small, slow moving glaciers since it is complex to obtain accurate horizontal velocities from remote sensing in such cases. Indeed, slow surface motion requests large time periods between two images to capture the displacement at a sufficient signal-to-noise ratio. During such longer time periods, visual (for optical data) and radar (for interferometry) coherence is often lost due to surface changes. However, it is easier to quantify surface elevation change and in turn emergence velocities after correcting for surface mass balance. Furthermore, it can be done for long time periods, up to several years, since the geodetic (DEM differencing) method is not sensitive to surface state. The accuracy of the emergence velocities method is good enough for the purpose of our paper but should be combined with horizontal velocity if a more precise localization of the frictional changes are intended. In the revised manuscript, we added details to the method and a new paragraph about method validation in the result uncertainties discussion (section 5.1).*

*Figures R1, R2 and R3 added to supplementary information.*

[Figure]

*Figure R3 - Basal friction coefficient inferred from horizontal velocity measurement (A) and from emergence velocity estimation (B).*

If I understand correctly, vertical velocities are derived from DEM differences (yielding dh/dt) and a mass balance model? If so, why is there a discussion of surface-normal velocities? Both dh/dt and b_dot are generally evaluated in the vertical direction, so there is no need for this? Intuitively, I'm surprised that this method works so well, but the results do look encouraging. But there should at least be some discussion of errors (which is missing for any of the results).

*Yes, you understood correctly. The quantity we derived is actually the vertical component of the surface-normal velocity, also called emergence (or subsidence) velocity. Vertical velocity would refer to vertical component of the velocity vector, which is a different quantity. We now only use the term "emergence velocity" to avoid confusion. Discussion of errors is extended in the revised manuscript (section 5.1) and new figures have been added in the supplementary material.*

2- There are conflicting assumptions in the paper that are not always discussed. For example, the derivation of bed topography is based on 'no sliding' (this is shortly discussed).

*The assumption on 'no sliding' for deriving bed topography is done for the glacier state in 2000 before the surging anomaly. Even with no flow instability going-on at this time, this is probably a rough assumption since we show that the two glaciers were temperate in the detachment area. However, this seems to provide good enough results when comparing with surface topography after the collapses. In the upper glacier parts, the no-sliding assumption is in agreement with the high friction inferred from inverse modeling. This assumption is now discussed in more detail.*

Friction parameters are derived from a linear sliding law, but the discussion is entirely in terms of a plastic till.

*Deriving friction parameters from a linear sliding law does not imply any choice of the physical processes behind the friction since the inversion is done at fixed times. The friction coefficient β can be expressed in the framework of plastic till theory which would give:*

$$\beta u_s = N \times tan(\phi) + c$$

*We do not think that inverting the friction using a linear sliding law is incoherent with a plastic till behavior since a linear sliding law can be viewed as a parametrization where the coefficient β includes the physics behind the processes taking place. However, in such approach, the value of β is only valid at the time for the inversion. For example, Minchew et al. (2016) used a linear sliding law to provide evidence of plastic behavior. We clarified this point now in the manuscript:*

*"The use of a linear friction law in our inversion is a parametrization where β includes these physics and is only valid at the time of the inversion. The change in friction coefficient β can here be therefore interpreted in terms of a plastic till."*

3-  I would like a bit more information about how stresses are divided between basal shear stress and lateral stress. In a valley geometry, the bed-parallel stress can be both lateral and basal.

*In our study, the force balance analysis is done on the part that detached. This means the existence of a western margin linking the detachment to the glacier body, which provides lateral stresses. See also our response in Technical Corrections. In the manuscript, lateral stress always refers to the stress within the part that detached and does not apply to the whole glacier. This has been clarified in the manuscript:*

*"The analysis of the dynamics and force-balance evolution on an area restricted to the detachment zone (dashed lines in Figure 8) reveals both similarities and differences between the two events (Figure 9). Further references to "lateral stress" apply to the detachment zone and not to the whole glacier. It refers to the stress provided by the shearing interface between the stable and the instable part of the glacier"*

4-  The Kolka Glacier case is interesting with a rock fall on it. There is a simple argument to be made that for a plastic till the addition of a mass on top of the glacier will lead towards instability if the glacier slope is larger than the friction angle of the till, without invoking pore water pressure changes. Is this potentially the case here?

*From satellite observation, we do not observe any evidence of external loading on the top of the glacier in the previous years. Such event would have been visible and is unlikely given that there is no steep rockwall overlooking the glaciers. However, bulging (due to surge-like behavior) associated to strong melting increase in the tongue area has steepened the glacier surface in the*

*tongue region (Kääb et al., 2018). This could also act as a trigger if the tongue surface slope would have reached the friction angle but changes in pore water pressure still have to be invoked to initiate the instability. We computed driving stress at different periods on the detachment from surface slope and thickness (**Error! Reference source not found.**9B of the manuscript). It shows an increase of 17% of the driving stresses that may have contribute to reach still strength. We mention it, in the manuscript.*

5- The abstract mentions that this is a response to recent increases of surface melting and rain. Neither is shown in the paper. This is an important conclusion and only enters the paper via a mass balance model that is discussed elsewhere. For such a substantiative statement it seems like there needs to be at least some amount of backup (e.g. a figure of temperature/precip changes)

*Yes, this result comes from the previous study of Kääb et al. (2018), a figure is already published in their supplementary material (Figure S10). We added a reference in the discussion (section 5.3).*

**Technical corrections**

I won't list grammatical issues, there are too many. This paper needs a very careful editorial revision.

*This has been done.*

Some other comments:

p.2, l.7: unique -> rare (it's not unique you mention another example in the next sentence...)

*Done*

p.3, l.10: are the two X-band images from the same time of year? Otherwise could the penetration depth change with snow wetness?

*The TanDEM-X images were acquired in June 2011, April 2013 and April 2014 (specified in Table 1). So there could be some penetration depth change between 2011 and 2013. However, ERA-interim reanalysis and Sentinel-1 backscatter images over the period 2015-2016 (warmer than 2011) show that no melt occured in the accumulation area (above 5800 m a.s.l.) before mid-June (see Figure R5). Changes in penetration depth are therefore likely not significant between June 2011 and April 2013. We modified the manuscript:*

*"The effect of uncertainty linked to radar penetration in the TanDEM-X data should be minimized when comparing same wavelength data (X-band) at similar times of the year. Change in penetration depth between the TanDEM-X data of 2011 (early June) and of 2013 (mid April) due to different snow wetness should be also limited because surface melting in the accumulation area of the Aru glaciers only occurs from around mid-June on (Kääb et al., 2018)."*

[Figure]

*Figure R4 - Elevation of the wet-dry snow transition in 2015 and 2016. Figure taken the supplementary material of (Kääb et al., 2018).*

eqn (2): d should be y

*Done*

p.6, l.2: which two cases?

*We mean here the cases of Aru-1 and Aru-2. This is now clarified in the manuscript.*

p.7, l.31/32: I don't understand that sentence at all (.. external side of the curve ..)

*The sentence has been clarified: « Along the left bank of the glacier, close to the terminus of Aru-1, shear stress is about 6-7 kPa and was not more than 15 kPa at the terminus. »*

p.10, l.2: sec 5.2 is a self-reference...

*We wanted actually refer to sec 5.3...  The manuscript has been corrected.*

p.10, l.10: How did you observe bedrock roughness. I thought this was all till covered?

*This is a qualitative statement based on field pictures (see Figure R5). You are right that we do not really observe the bedrock interface but rather the failure plan, which can be different. However, this provides evidence for a rather smooth interface between the glacier and its substrate. Manuscript has been updated.*

[Figure]

*Figure R5 – View of Aru-2 after collapse (November 2016, Picture: T. Yao).*

p.10, l.17/18: The MacAyeal and Tsai references don't quite seem appropriate here; they don't show plastic till, they assume it in their models.

*Yes, we removed these two references.*

p.10, last paragraph: I find some of the discussion here confusing. What do you mean when you state that "plastic rheology becomes the only source of resisting forces"? Or "increasing pore pressure ... quickly reduced basal shear stress"? Increasing pore water pressure reduces effective stress (not shear stress) and through that the strength of the till. In a plastic rheology you can't reduce the shear stress to the strength; till strength is a limiting stress.

*We mean that when the failure occurred at the detachment margin and lateral stresses are not able anymore to contribute to force balance, the only source of resisting force became the basal friction, which behaves plastically when till strength is reached. In other word, as you say, the resisting stress became limited to the till strength which is not able to compensate driving stress anymore, leading to collapse.*

*Yes we exactly mean what you wrote here. Increasing pore water decrease effective normal stress and therefore till strength. Because till strength is the limiting stress and driving stress is superior to this limit, basal shear stress actually takes the till strength value and locally decreases as the till strength decreases.*
*We clarified this paragraph in the reviewed manuscript.*

p.11, l.17: What would cause higher lateral stresses? See also my earlier comment: when does a basal stress become a lateral stress in a valley geometry?

*In our study, we refer to lateral stress at the detachment margin and not over the whole glacier. Therefore, our force balance analysis is done on the part of the glacier that detached. On this delimited area, lateral stresses exist along the western glacier margin where the detachment is connected to the glacier body over a significant thickness of ice. In particular, the ice body, west of the detachment, is likely cold based and provides significant lateral resistance to flow in the detachment area. In this case, higher lateral stress would be caused by greater ice thickness or more likely by less damaged ice at the detachment margin, confirmed by the much less developed crevassed area on Aru-2. The manuscript has been clarified on this point. See also response to specific comment no 3.*

*References:*

*Minchew, B., Simons, M., Björnsson, H., Pálsson, F., Morlighem, M., Seroussi, H., Larour, E. and Hensley, S.: Plastic bed beneath Hofsjökull Ice Cap, central Iceland, and the sensitivity of ice flow to surface meltwater flux, Journal of Glaciology, 62(231), 147–158, doi:10.1017/jog.2016.26, 2016.*

- **Reviewer 2 (I. Rogozhina)**

This paper presents a model-based interpretation of temporal changes in the internal dynamics, basal friction and stress states of two glaciers in western Tibet to explain their catastrophic collapses in 2016. The inversion method used to derive glacier model results is rather unusual in this I agree with reviewer 1 – but it seems to yield rather good results. I still think that this method should be validated on a glacier that has measurements of horizontal velocities, vertical surface changes and ideally bedrock topography to make the case that it is operational. There are quite some examples of such glaciers, especially in the European Alps. Inversion methods can be quite tricky, since they derive whatever one wants to obtain, especially when multiple parameters are estimated in parallel. Nevertheless, this study is an impressive contribution to the state-of-the-art understanding of the glacier dynamics and addresses the challenge of the glacier model initialization in a neat manner, even though it has a significant overlap in terms of motivation and conclusions with the paper featuring the same authors (Kääb et al., 2018). The language is quite remarkable, as reviewer 1 has pointed out, and I am rather surprised to see so many experienced co-authors – including native speakers who do not seem to have read the paper. With this review I encourage them to have a look at it. I believe that this paper will merit publication in TC after moderate revisions.

**MAJOR POINTS:**

1. As I mentioned in my summary, the authors should prove that their inversion method is operational by validating it on a glacier with more measurements (see above).

*We compare the results obtained by our method (2013-2014) with a more standard inversion method based on horizontal velocity measured in December 2013 by TerraSAR-X offset tracking. This shows good agreement and validates our inversion method. See also response to reviewer 1 (specific comment no 1). A paragraph has been added in the main manuscript as method validation (section 5.1) referring to new figures in the supplementary material.*

2. I absolutely agree with reviewer 1: All the points he has raised are valid and I am looking forward to seeing responses to his concerns. In addition, I feel that his specific point 2 needs further exploration: It would be worth looking at how friction angle in equation 4 (in addition to friction coefficients) changes over time leading to the collapses of the glaciers. In addition to showing how the subglacial till changed its properties in response to warming and increased meltwater supply, this experiment will provide an estimate of the yield stress needed to enable such a failure. A very useful exercise for the future diagnostic experiments that will empower predictions of similar glacier failures and an important exercise to support the conclusions of this study.

*See response to reviewer 1. Concerning friction angle: this is unfortunately not possible since we do not know the water pressure and effective normal stress. The friction angle is actually a constant through time that depends on the material property. The yield stress is then only a function of the effective normal stress. Assuming a typical value of friction angle and that basal shear stress reached the yield stress in most of the detachment, we could infer water pressure from equation 4. This would have to be analyzed by additional investigations based on subglacial hydrological models to be relevant. This is beyond the scope of the paper but inferring subglacial water pressure would be clearly a valuable next step of studying the Aru collapse.*

3. I don't believe much in the climate forcing provided by ERA-Interim in such high-topography, steep-gradient environment, especially after I learned from this paper that the precipitation rate had to be multiplied by a factor of 4. Could the authors compare ERA-interim fields with High Asia Refined (HAR) analysis (Maussion et al., 2014)? I suggest that the authors perform sensitivity tests to assess the uncertainties in their results coming from the mass balance estimates using HAR.

*In our study, surface mass balance modeling is only used to compute surface-normal velocities from elevation change measurement and infer basal friction. We assessed the sensitivity of our results to surface mass balance by inverting basal friction during the period 2013-2014 under different surface mass balance reconstructions: (i) No surface mass balance correction, (ii) Modeled surface mass balance divided by two and (iii) Modeled surface mass balance multiplied by two. These hypotheses are rough and introduce greater variation in the surface mass balance than the actual uncertainty on this reconstruction constrained by satellite measurement and field data (see (Kääb et al., 2018), sup. mat.). The friction field variability presented in Figure R7 is therefore a conservatively large estimate of the uncertainty introduced by the surface mass balance. It shows that friction in the detachment area where surface-normal velocities are high,*

*due to unbalanced geometry, is not very sensitive to surface mass balance reconstruction. This makes our results reliable in this glacier part, which is the focus of the study.*

[Figure]

*Figure R6 – Friction coefficient β inferred for different surface mass balance modifications on Aru-1 between 2013 and 2014. Absolute values.*

*This figure is now included in the Supplement of the manuscript.*

*The ability of ERA-interim reanalysis to reproduce observed mass balance has been already compared to MERRA-2 and HAR reanalysis in (Kääb et al., 2018). ERA-interim provides the best mass balance agreement with observations compared to the two other products. This is mainly due to a sudden increase of precipitation in the mid-nineties that is not captured by MERRA-2 (see Figure R8). HAR reanalysis gives an unrealistic trend in precipitation that makes mass balance modeling not able to reproduce the observations. However, the mean values of precipitation provided by HAR are in good accordance with our corrected ERA-interim precipitation (factor 4) (see Figure R9). The factor 4 is also confirmed by high elevation AWS measurement in the region (200 km away). Please refer to the sup. mat. of Kääb et al. (2018) for greater details.*

[Figure]

*Figure R7 – Left :Annual precipitation at the different weather stations and from reanalysis at the location of the Aru range (see Kääb et al. (2018)). Right :Annual temperature anomaly relative to the 1980-2000 mean at the different weather stations and from reanalysis at the location of the Aru range (see Kääb et al. (2018)).*

[Figure]

*Figure R8 - Corrected ERA-interim annual precipitation (red line) using glacier geodetic mass balance observations compared with the 10km resolution HAR reanalysis (blue line) (see Kääb et al. (2018)).*

**MINOR POINTS:**

The methods section is sloppy. For example, I am missing a table with model parameters. In general, the methods have to be more detailed. This is not a Nature paper, there is space for the description of methods. The supplementary materials are no accessible through the online system. There are some citations of materials in the supplement, which I cannot access.

*We did not find any issues to access the supplement through the article webpage; this should be accessible to all. We made the choice to not describe the model in details and focus on the application of it since both mass balance and thermo-mechanical model are already described in other published studies (Gilbert et al., 2014, 2016; Kääb et al., 2018). In the revised manuscript we added a table with model parameters and described more the methods (see also response to Major point 1).*

Page 7, lines 13 – 14: Cannot it be influenced by a larger error in the bedrock estimate?

*This is unlikely since bedrock topography is quite well constrain in the detachment area by the post-collapse Pleiades DEM. Also high friction on the tongue of Aru-2 is confirmed by the fact that the Aru-2 front never advanced before the collapse contrary to the Aru-1 front. We added a sentence in section 5.1 about uncertainty. "Bedrock topography is well constrained in this part from the post-collapse Pléiades DEM, giving additional confidence in the friction reconstruction over the detachment area.*

Page 9, line 9 – 13: This requires a proof.

*See response to major point no 3. An additional figure has been added in the supplementary information.*

Page 9, line 27: "from temperate to cold basal conditions" - the other way around?

*Yes. The manuscript has been corrected.*

Page 10, section 5.3: The field data are only available for Aru 1. Are the authors sure that Aru 2's bed has the same lithology?

*This is shown by multi-spectral lithological analysis from satellite images. See (Kääb et al., 2018).*

Page 11, lines 4 – 5: Any evidence from the little ice age glacial moraines to support this statement?

*No we did not find any. But melt rates are already very low in this kind of cold/dry environment. It makes relative melt rate increase significant in a context of climate warming as soon as a melt threshold is reached; especially in the accumulation area where water likely entered the glacier body (crevassed area, warmer ice temperature due to firn). Considering the current melt rate in the accumulation area (< 0.25 m w.eq. yr$^{-1}$, see Figure R10), it is likely that no melt occurred at similar elevation during the Little Ice Age.*

[Figure]

*Figure R9 – Modeled annual melt and rain at 5850 m a.s.l. (Kääb et al., 2018)*

**References:**

Kääb, A., Leinss, S., Gilbert, A., Bühler, Y., Gascoin, S., Evans, S. G., Bartelt, P., Berthier, E., Brun, F., Chao, W.-A., Farinotti, D., Gimbert, F., Guo, W., Huggel, C., Kargel, J. S., Leonard, G. J., Tian, L., Treichler, D. and Yao, T.: Massive collapse of two glaciers in western Tibet in 2016 after surge-like instability, Nat. Geosci., 11(2), 114–120, doi:10.1038/s41561-017-0039- 7, 2018.

Maussion, F., D. Scherer, T. Mölg, E. Collier, J. Curio, and R. Finkelnburg: Precipitation seasonality and variability over the Tibetan Plateau as resolved by the High Asia Reanalysis, J. Climate, 27, 1910-1927, doi:10.1175/JCLI-D-13-00282.1, 2014.